# Impact of C-terminal amino acid composition on protein expression in bacteria

Marc Weber[1] (ID), Raul Burgos[1], Eva Yus[1], Jae-Seong Yang[1], Maria Lluch-Senar[1] (ID) & Luis Serrano[1,2,3,*] (ID)

## Abstract

The C-terminal sequence of a protein is involved in processes such as efficiency of translation termination and protein degradation. However, the general relationship between features of this C-terminal sequence and levels of protein expression remains unknown. Here, we identified C-terminal amino acid biases that are ubiquitous across the bacterial taxonomy (1,582 genomes). We showed that the frequency is higher for positively charged amino acids (lysine, arginine), while hydrophobic amino acids and threonine are lower. We then studied the impact of C-terminal composition on protein levels in a library of *Mycoplasma pneumoniae* mutants, covering all possible combinations of the two last codons. We found that charged and polar residues, in particular lysine, led to higher expression, while hydrophobic and aromatic residues led to lower expression, with a difference in protein levels up to fourfold. We further showed that modulation of protein degradation rate could be one of the main mechanisms driving these differences. Our results demonstrate that the identity of the last amino acids has a strong influence on protein expression levels.

**Keywords** bacteria; bias; C-terminal; degradation; expression
**Subject Categories** Computational Biology; Translation & Protein Quality
**Mol Syst Biol. (2020) 16: e9208**

## Introduction

Protein sequence is shaped by many evolutionary constraints, acting at different levels of the gene expression process. Identifying which sequence features determine the efficiency and accuracy of protein expression has been the subject of intense research. Sequence variations previously believed to be neutral, such as the choice of synonymous codons, have been shown to be under selection (Hanson & Coller, 2018), revealing new mechanisms interacting with the translation process. Most studies have focused on the region close to the N-terminal, where some of the most important mechanisms of translation initiation occur (Goodman *et al*, 2013; Reeve *et al*, 2014; Espah Borujeni & Salis, 2016).

Much less is known, however, about the potential evolutionary pressures acting at the C-terminal, apart from basic protein function and structure. Early studies showed a differential preference for specific codons upstream of the stop codon in *Escherichia coli* (Brown *et al*, 1990; Arkov *et al*, 1993; Björnsson *et al*, 1996; Berezovsky *et al*, 1999) and *Bacillus subtilis* (Rocha *et al*, 1999; Palenchar, 2008) proteins. The properties of the last two amino acids were shown to modulate the efficiency of translation termination at the UGA stop codon context in *E. coli* (Björnsson *et al*, 1996), in particular for highly expressed genes. Also, several C-terminal sequence motifs were found to induce stalling of translation termination (Hayes *et al*, 2002; Woolstenhulme *et al*, 2013). Besides, degradation signals have been identified at the C-terminal of proteins in a variety of bacterial species (Sauer & Baker, 2011), such as the ssrA tag which targets proteins to the ClpXP protease in *E. coli* (Gottesman *et al*, 1998). Thus, changes in the efficiency of translation termination and recognition of the C-terminal region by the protein degradation machinery are two potential mechanisms that could drive preferences in the C-terminal composition of proteins.

Translation is one of the most energy-intensive processes in the cell, consuming about 40% of the cellular energetic resources in fast-growing bacteria (Russell & Cook, 1995). Thus, sequence features at the C-terminal that lead to variations in protein abundance, by modulating either translation or degradation rates, are likely to be under selection. However, the impact of C-terminal composition on protein abundance remains largely unknown. Many studies have identified sequence features that influence translation efficiency, but most of them have focused on the 5′ end region or the bulk of the coding sequence (Kudla *et al*, 2009; Goodman *et al*, 2013; Cambray *et al*, 2018). In these studies, the use of synthetic libraries of a reporter gene with randomized sequence has proved as a robust approach to evaluate the functional impact of sequence variation. A similar approach applied to variations in the C-terminal region would provide useful information to identify sequence features associated with higher or lower protein abundance.

Here, we investigated the impact of C-terminal sequence composition on protein expression. Firstly, we leveraged the considerable increase in the past decade in the number of available bacterial genome sequences (Loman & Pallen, 2015) to study C-terminal composition biases of 1,582 genomes across the bacterial taxonomy.

---

1 Centre for Genomic Regulation (CRG), The Barcelona Institute of Science and Technology, Barcelona, Spain
2 Universitat Pompeu Fabra (UPF), Barcelona, Spain
3 ICREA, Barcelona, Spain
*Corresponding author. Tel: +34 933160101; E-mail: luis.serrano@crg.eu

Such a large-scale comparative analysis allowed us to reveal the universality of the sequence compositional patterns, as well as preferences which seem to be species-specific, and unveil their association with protein function and protein abundance. Secondly, we experimentally assessed, in a high-throughput manner, the influence of C-terminal composition on protein expression levels in the model organism *Mycoplasma pneumoniae* using the ELM-seq technique (Yus *et al*, 2017). We built a random library of the *dam* reporter gene with varying C-terminal sequence, covering all possible combinations of the last two codons and the six nucleotides following the stop codon. By measuring the expression levels of all variants, we showed that the identity of the last two amino acids has a strong impact on protein abundance. We validated these results by varying the last residue of a different protein in the same species. Furthermore, we provide evidence associating the identity of the last C-terminal amino acid with protein degradation rate. Overall, our results show that in bacteria, the C-terminal residue of protein sequences modulates protein expression levels and is under selective pressure.

## Results

### Analyzing C-terminal compositional biases in bacteria

#### C-terminal amino acid and codon composition in bacteria is biased

We investigated biases in codon and amino acid composition of the C-terminal region of bacterial protein sequences. We retrieved all protein sequences from the RefSeq database (Haft *et al*, 2018), using the reference and representative genome collections, in order to achieve a broad coverage of bacterial species across taxonomy. To avoid over-representation of duplicated proteins within the same bacterial species, we removed proteins which presented both a high overall sequence identity and a high identity of their C-terminal region (see Materials and Methods). We obtained a database of approximately 4.8 M protein sequences covering 1,582 genomes and 1,516 species, which we used as a starting point for all the following analyses.

When studying all species in the bacterial kingdom, we found that the amino acid composition at the last position upstream of the stop codon differed significantly from the bulk amino acid composition (Fig 1A). In particular, positively charged amino acids were enriched at position −1, with the frequency of lysine and arginine being 2.32 times (two-sided Fisher's exact test $P \approx 0$ within numerical error) and 1.76 times ($P = 7.7\text{e-}30$) higher, respectively. In contrast, the occurrence of threonine was 2.25 times ($P = 2.2\text{e-}308$), and that of methionine, 2.02 times ($P = 7.1\text{e-}51$) lower. Due to the large number of sequences considered in this analysis, all biases were significant with extremely small $P$-values, even after correcting for multiple testing. Interestingly, we observed a gradient in the intensity of the biases for all amino acids that showed a statistical difference at the C-terminal, except for threonine that was specifically depleted at the −1 position. This gradient was more evident in the case of arginine and lysine, whose biases decreased from a maximum value at the C-terminal position toward the bulk, with an odds ratio still significant for lysine of 1.16 ($P = 2.3\text{e-}23$) at position −20. All hydrophobic amino acids, except phenylalanine, were found to be disfavored at the last position, with fold changes in

frequency ranging from 0.49 to 0.87 times ($P = 7.5\text{e-}6$). The amino acid frequencies detected at positions −1 and −2 differed from those found in disordered regions in proteins, indicating that the preferences observed are not due to the C-terminal being in general unstructured in proteins (Kleppe & Bornberg-Bauer, 2019) ($\chi^2$ test $P \approx 0$, Appendix Fig S1).

In order to explore possible cooperative effects, we investigated the frequency of amino acid pairs at the last two positions from two points of view. First, we compared the frequency of the C-terminal amino acid pair to the frequency of the same dipeptide in the bulk (Fig 1B), thus correcting for dipeptide biases observed in proteins (Gutman & Hatfield, 1989). Pairs of positively charged amino acids were found enriched, and pairs of hydrophobic amino acids depleted, recapitulating the biases observed for individual amino acids at the last two positions. Second, we compared the frequency of an amino acid pair at the C-terminal to its expected frequency under the assumption that positions −1 and −2 are independent (null model) (Fig 1C). The deviation from the expected frequency, or epistasis, revealed that many of the pairs of repeated residues were more frequent than expected, in particular, CC-Stop (odds ratio 5.16), MM-Stop (odds ratio 2.93), WW-Stop (odds ratio 2.24), HH-Stop (odds ratio 1.84), and KK-Stop (odds ratio 1.80) (binomial test, $P \approx 0$ for all). These five amino acid pairs also exhibited a positive epistasis in the bulk (Appendix Fig S2) (odds ratio 1.29–1.53), suggesting that part of the observed positive interaction was not specific to the C-terminal. In addition, because cysteine, methionine, tryptophan, and histidine were also the least frequent amino acids in general, a small number of proteins with a conserved functional motif that include those dipeptides could easily lead to the over-representation of the pair. In particular, the CC pair presented the strongest positive epistasis effect. In the family of metal sensor proteins, binding to metal ions is often mediated by multiple cysteine thiolates (Rosen, 1999; Osman & Cavet, 2010). While the metalloregulatory protein families are diverse in structure and functions, some of them possess a conserved cysteine-rich motif close to the C-terminal (Ma *et al*, 2009). Indeed, at least 396 out of the 2,034 proteins in our database that possess a CC-Stop motif belonged to orthogroups related to metalloprotein families. On the opposite, the amino acid pairs that exhibited the most negative epistasis were some of the XP-Stop combinations DP-Stop (odds ratio 0.33), GP-Stop (odds ratio 0.43), PP-Stop (odds ratio 0.44), and FP-Stop (odds ratio 0.54) (binomial test, $P \approx 0$ for all). Two of these dipeptides (DP and PP) were previously shown to induce the strongest level of translation stalling and tagging by the ssrA ribosome rescue system in *E. coli* (Hayes *et al*, 2002), providing a possible explanation for this negative selection.

In order to explore whether composition biases are conserved across the bacterial phylogeny, we grouped bacterial genomes into taxonomic clades at the level of phyla, and analyzed composition biases of sequences from each reduced set of bacterial species (Fig 1D). Overall, the main biases at the C-terminal position were present in virtually all phyla. Interestingly, proline was found to be enriched in 12 phyla and depleted in other 16 phyla (e.g., 0.23 odds ratio in Tenericutes, 0.17 in Fusobacteria). The same analysis at a finer level of the taxonomy (Appendix Figs S3 and S4) showed that the biases for proline varied greatly across taxonomic clades. Interestingly, the biases across phyla for threonine anticorrelated with the biases for lysine (Pearson $r = -0.84$).

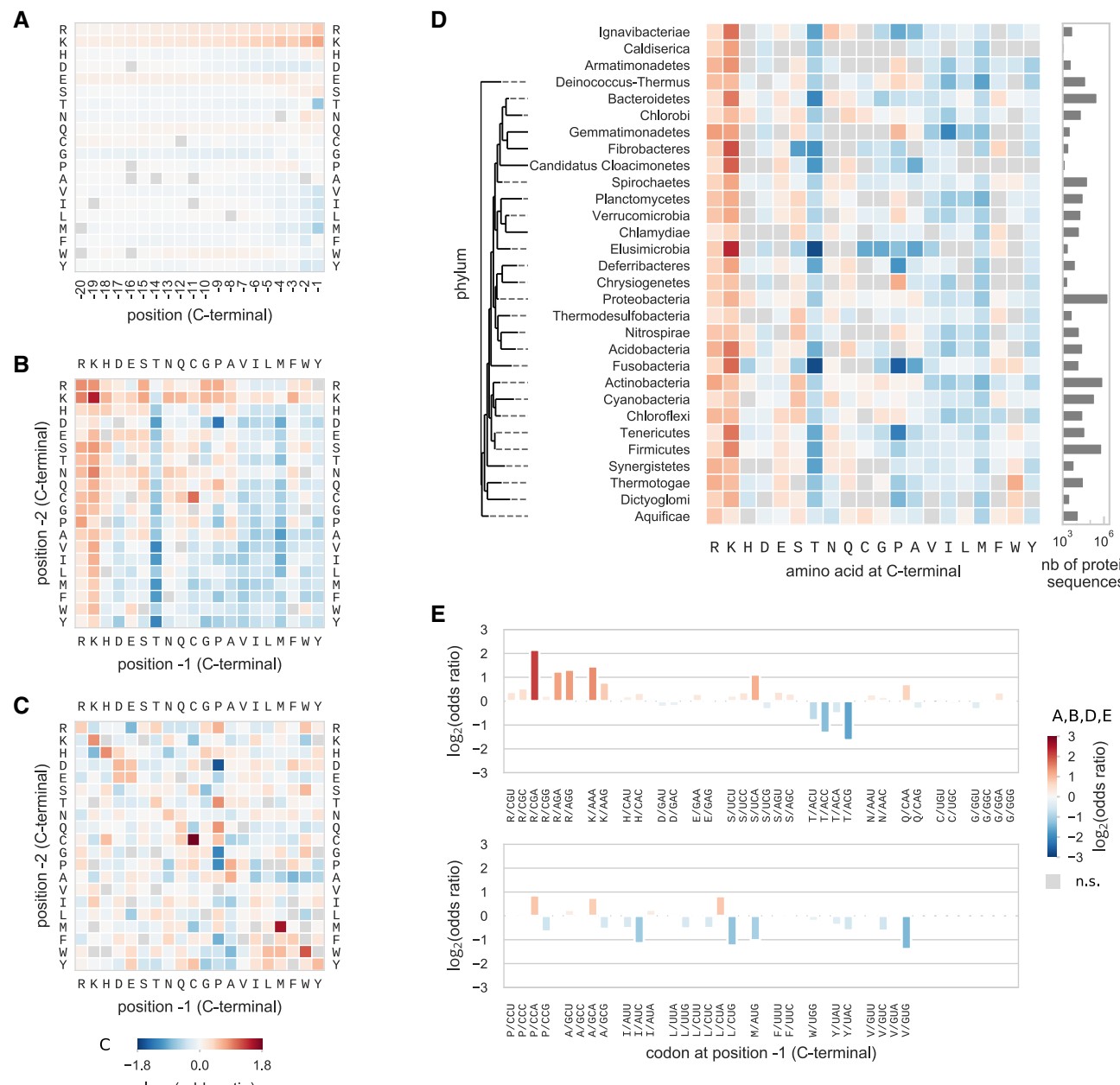

**Figure 1. Biases in C-terminal protein sequence composition in the bacterial kingdom.**

Amino acid and codon composition at the C-terminal of bacterial protein sequences shows higher (red) or lower (blue) frequency when compared to their frequency in the bulk of sequences (same color code for panels A, B, D, and E). Significance of the biases was tested using Fisher's exact test and multiple testing correction with 5% false discovery rate within each plot category.

A   Position-specific amino acid composition biases for the last 20 amino acids at C-terminal.

B   Composition bias of the last two amino acids compared to the frequency of the di-amino acid in the bulk.

C   Epistasis between the last two amino acids. Frequency of the pair was compared to the expected frequency if the two positions were independent. In this case, significance was tested with the binomial test and the same multiple testing correction. Color code has a smaller range than in other panels.

D   Amino acid composition bias at the position −1 for individual phyla. Phyla were ordered following an approximate phylogenetic tree.

E   Codon composition bias at the last position.

### Pattern of C-terminal codon biases and its relationship to the stop codon context

We then asked whether those biases were restricted to specific codons or were rather independent of the identity of the synonymous codon (Fig 1E, and Appendix Figs S5 and S6). We found that both codons encoding for lysine (AAA and AAG) were enriched at the −1 position at fairly similar levels (odds ratios 3.13 and 2.43). Similarly, all four codons coding for threonine were

depleted, despite some variations in their odds ratios. Interestingly, only three (CGA, AGA, and AGG) out of the six arginine codons were strongly enriched, with the highest odds ratio for the CGA codon (4.44). In the group of hydrophobic amino acids that were found to be depleted (valine, isoleucine, leucine, methionine, tyrosine, and tryptophan), some synonymous codons were more strongly underrepresented, although in general, the majority of these codons were disfavored (11 out of 17). We emphasize that these preferences were not related to differences in synonymous codon usage, since all biases were computed with respect to the frequency of codons in the bulk. Interestingly enough, with the exception of arginine AGA and AGG codons and lysine codons, in all other amino acids there was a strong preference for an A at the third position.

We reasoned that some of these variations in C-terminal codon biases could be related to specific interactions with the stop codon. Thus, we analyzed the biases in C-terminal codon composition in each of the three stop codon contexts, both at the level of bacterial kingdom (Fig 2) and at the level of individual phyla (Appendix Fig S7). Overall, codons that presented strong enrichment/depletion were enriched/depleted in all stop codon contexts. However, some variations in codon biases were also observed. We found that the identity of the last base of the C-terminal codon had an influence on codon biases depending on the stop codon context (Fig 2, Appendix Fig S8). In particular, NNA codons were more often enriched than other codons in all three stop codon contexts (distribution of the log2(odds ratio), independent $t$-test $P = 4e-24$). This preference was exacerbated in the UGA context, where NNA codons were clearly favored over NNG codons ($P = 1.8e-49$). Interestingly, the presence of a C-terminal codon ending with an A in the context of a UGA stop codon creates an overlapping starting AUG codon, while the other bases will result in the less favored UUG, GUG, CUG start codons. In bacteria, genes that overlap on the same polycistronic mRNA are common [Johnson & Chisholm, 2004; 34% in *E. coli* MG1655 (Keseler *et al*, 2013; Tian & Salis, 2015)]. Both genomic compression and translational coupling are believed to promote short overlaps of 4 nt, in which the use of the UGA stop codon is particularly frequent (Lillo & Krakauer, 2007). Thus, we reasoned that the preference for NNA codons in the UGA stop codon context could reflect selection constraints of overlapping start codons. In order to test this hypothesis, we analyzed (Fig 2) the codon biases in the UGA stop codon context when excluding genes for which the start codon of the downstream gene overlapped at nucleotide position $-1$, e.g., NNA-UGA where AUG is the start codon. In this case, the preference for NNA codons was greatly reduced (Appendix Fig S8D, $P = 0.12$), suggesting that this effect is mainly due to the overlapping of start codons.

### Pattern of C-terminal amino acid biases is qualitatively independent of functional category

Next, we asked whether the biases in C-terminal amino acid composition are specific to some protein functional classes. We hypothesized that some of the biases could be driven by a small group of proteins with amino acid composition different from the average. In order to explore this hypothesis, we analyzed protein functional groups in two ways.

First, we studied whether the enrichment for lysine and arginine at the C-terminal region could be due specifically to transmembrane proteins. It has been suggested that the orientation of transmembrane domains is determined by the enrichment of positively charged residues in cytoplasmic loops, rather than in periplasmic loops, a mechanism known as the positive-inside rule (Driessen & Nouwen, 2008; Dalbey *et al*, 2011). Thus, transmembrane proteins whose C-terminal domain is cytoplasmic might present an enrichment of lysine and arginine compared to their frequency in the bulk of the sequence. Indeed, positive charges have been shown to be enriched in the C-terminal cytosolic region of transmembrane proteins in *E. coli* (Charneski & Hurst, 2014). We classified proteins as membrane or cytoplasmic based on computationally predicted localization for a selection of 364 bacterial species, and computed the C-terminal amino acid composition biases for each of the two classes (Fig EV1). Positively charged residues were found to be strongly enriched in the last 10 positions of the C-terminal of membrane proteins (mean odds ratio K, 2.10, R, 1.69). The same biases were weaker for cytoplasmic proteins (mean odds ratio K, 1.57, R, 1.22). Hydrophobic amino acids were found to be depleted in both protein categories, although more strongly in membrane proteins (mean odds ratio for A, V, I, L, M, F, W, Y, 0.72 for membrane, 0.84 for cytoplasmic). Apart from these differences, we found a similar pattern of biases at position $-1$ for membrane and cytoplasmic proteins (Fig EV1C), including depletion of threonine, methionine and hydrophobic residues, and enrichment of lysine and arginine. Thus, while membrane proteins have a higher frequency of positively charged residues at the C-terminal, they only partially contribute to the global amino acid composition biases observed at the level of all proteins.

Second, we systematically classified proteins into functional categories by assigning each protein sequence to a Cluster of Orthologous Groups (COG) category. We computed the composition biases within each of the 23 functional categories, by comparing the frequency of amino acids at the C-terminal to the bulk frequency of sequences in the same category (Fig EV2). The previously observed general biases were qualitatively maintained in the vast majority of the functional categories. Importantly, the overall pattern of biases was maintained despite differences in the bulk frequency of some amino acids between categories. For example, ribosomal proteins contain many positively charged residues that are essential for their interaction with RNA (Klein *et al*, 2004), and as a consequence, proteins in the J category "Translation, ribosomal structure and biogenesis" have a higher frequency of lysine in the bulk (6.01% compared to 4.23% in average for the other categories). However, in this case, the enrichment for lysine at the C-terminal still holds when compared to its frequency in the bulk (16.2% compared to 6.01%, odds ratio 3.02). Similarly, in each of the other 22 categories, the frequency of lysine was higher than in the bulk.

Therefore, the main pattern of amino acid biases observed at the last position of the C-terminal is qualitatively independent of functional category.

### C-terminal amino acid identity is associated with protein abundance

One possibility is that the observed C-terminal biases could be driven by an underlying mechanism affecting protein abundance, such as translation termination efficiency or protein degradation. If this is the case, we would expect to observe differential biases for proteins that are highly abundant than for lowly abundant proteins. We examined the association between protein abundance and

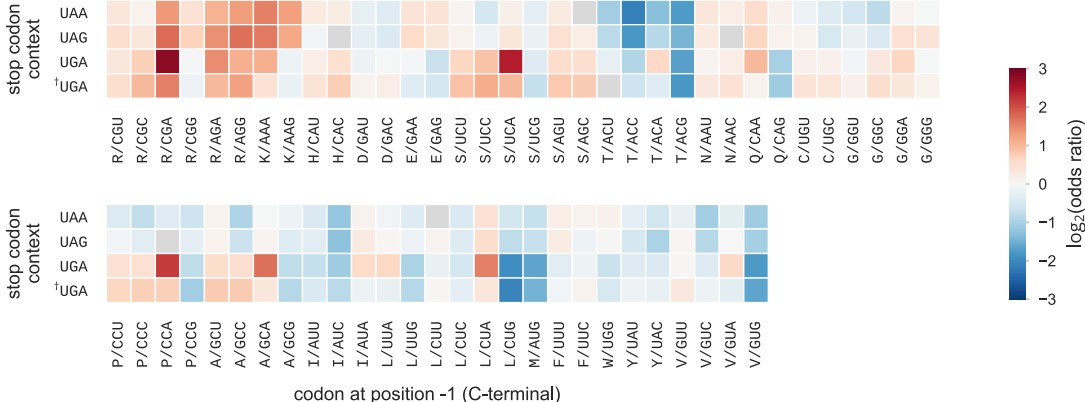

**Figure 2. Codon composition biases at C-terminal and stop codon context.**

In the bacterial kingdom, protein sequences were first classified by their stop codon context: UAA, UAG, or UGA. In the case of the UGA stop codon context, genes were excluded (†UGA context) for which the start codon of the downstream gene was overlapping with the stop codon at nucleotide position −1, e.g., NN**A-UG**A where AUG is the downstream start codon. Codon frequency at C-terminal was then compared to the bulk codon frequency within each stop codon context class. Significance of the biases was tested using Fisher's exact test and multiple testing correction with 5% false discovery rate within each class (all cases were significant).

C-terminal biases in 13 bacterial species for which at least 40% of the proteins had experimental abundance value from the PaxDB database (Wang *et al*, 2015). We categorized proteins into low (percentiles 0–20), medium (percentiles 20–80), and high (percentiles 80–100) abundance, and computed the amino acid composition biases in each group (Fig 3). C-terminal amino acid frequencies in each group were compared to the bulk amino acid usage of the proteins in the same group (null), such that differences in bulk amino acid usage between lowly and highly abundant proteins (Appendix Fig S9) were corrected for. Highly abundant proteins showed a stronger enrichment at the C-terminal (position −1) of K but not for R, and depletion of T, P, and C compared to lowly abundant proteins. In addition, amino acid biases at position −3, with the exception of cysteine, were very similar between the two abundance categories. Thus, the identity of the C-terminal amino acid at position −1 could be in part related to protein abundance.

### Pattern of amino acid substitution rates suggests C-terminal-specific purifying and positive selections

We then investigated whether evolutionary forces specific to the C-terminal position could be identified. We hypothesized that if the identity of the C-terminal amino acid can have an impact on protein expression and potentially on fitness, the pattern of amino acid substitutions at the C-terminal position should be significantly different from the pattern observed at other positions. More precisely, if the ancestral state is a favorable amino acid, purifying selection would decrease the substitution rate to non-favorable amino acids. Contrariwise, if the ancestral state is a non-favorable amino acid, positive selection would increase the substitution rate to favorable amino acids.

We analyzed 57 triplets of closely related bacterial genomes (Table EV1) with unambiguous phylogeny from the ATGC database (Kristensen *et al*, 2017). Using the maximum parsimony principle, we reconstructed the ancestral state and computed amino acid substitution frequencies at each position from the C-terminal (Fig 4A). The total frequency of all substitutions clearly increased from the positions in the bulk toward the C-terminal (Fig 4B), which likely

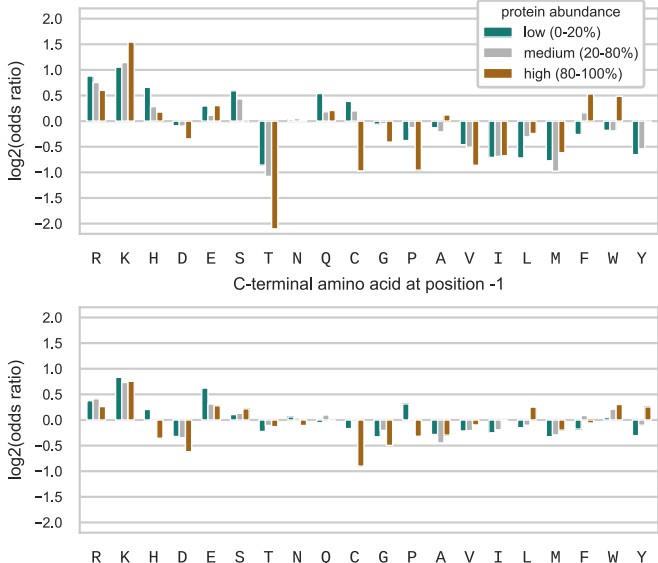

**Figure 3. Protein abundance and C-terminal amino acid identity.**

Proteins were categorized into low (percentiles 0–20, green), medium (percentiles 20–80, gray), and high (percentiles 80–100, brown) abundance, and C-terminal amino acid composition biases, with respect to the bulk frequency of each category, at position −1 (upper plot) and position −3 (lower plot), were analyzed for each of the three categories.

indicated a relaxation of purifying selection, as previously reported in eukaryotes (Ridout *et al*, 2010). One of the strongest determinants of the variation of evolutionary rates across sites is solvent accessibility (Echave *et al*, 2016). As residues at the C-terminal are usually unstructured and/or at protein surface (Jacob & Unger, 2007; Kleppe & Bornberg-Bauer, 2019), they can be expected to have a faster evolutionary rate than residues in the core of the proteins. We reasoned that the last two C-terminal positions have similar structural constraints and that comparing the substitution rates at the −1

position to the same rates at the −2 position would reveal selection forces specific to the C-terminal −1 position.

In order to allow reliable statistics, amino acids were classified into the following groups: positively charged (K, R), hydrophobic (A, I, L, M, F, W, Y, V), threonine (T), and others (H, D, E, N, Q, S, P, C, G). By comparing the substitution rates between groups at different positions from the C-terminal (Fig EV3), significant differences appeared between the last two positions (Fig 4, panels C and D). The substitution rate from hydrophobic amino acids to positively charged amino acids was significantly higher at the position −1 compared to the same rate at position −2 (two-sided Fisher's test, $P = 6.2e{-}05$, 1.86-fold difference). All substitution rates from any of the three groups to threonine were significantly lower at position −1 compared to position −2 (two-sided Fisher's test, $P = 6.2e{-}05$, 0.49- to 0.59-fold differences). Thus, the observed differences in the pattern of amino acid substitution rates at the position −1 (Fig 4D) suggest positive selection for positively charged amino acids and purifying selection against threonine that are specific to the C-terminal position.

## C-terminal amino acid identity impacts protein expression levels in *Mycoplasma pneumoniae*

As described above, the identity of the C-terminal residues could be related to protein abundance and be under selection in highly abundant proteins, suggesting that it could play a role in determining protein expression levels. To see whether this could be the case, we selected a genome-reduced organism with a simplified degradation (40) and translation (39) machinery, *M. pneumoniae*, to remove as many confounding factors as possible. In particular, it has only one release factor (RF1) recognizing the UAA and UAG stop codons (Grosjean *et al*, 2014), with the UGA codon coding for tryptophan. In addition, it lacks a peptidoglycan sacculus and no carboxypeptidase has been identified in this species (Wodke *et al*, 2015). The C-terminal amino acid biases in this species and in the taxonomic class it belongs to, Mollicutes, are representative of the pattern of biases observed in bacteria (Appendix Fig S10). We designed an experimental assay to measure, in a high-throughput manner, the expression levels of the *dam* reporter gene (DNA adenine methylase from *E. coli*) with a variable C-terminal region, using the ELM-seq method (Yus *et al*, 2017) (see Fig 5A and Materials and Methods). In this method, the relative methylation level of target sites on the DNA is measured by ultra-sequencing and has been shown to be approximately linear to the Dam protein abundance (Yus *et al*, 2017). In our study, we assume that the C-terminal extensions do not affect Dam activity, such that this relation remains valid. We built a random library of the *dam* reporter gene where six random nucleotides were added at the C-terminal of the Dam wild-type sequence, such that all combinations of the last two codons were covered. Six random nucleotides were also added downstream of the UAA stop codon, in order to study their influence, if any. Transcription of the *dam* gene was driven by either a strong or a weak promoter, resulting in two different average levels of expression.

We first examined the results in the weak promoter library. We analyzed the expression levels (DAMRatio) at the level of individual nucleotide identity for each of the random positions (Fig 5B). The nucleotide at the first two positions of each codon had the strongest influence, while the 3$^{rd}$ nucleotide position had little influence,

which suggests a similar expression level among synonymous codons. As expected, the nucleotide composition downstream of the stop codon had no influence on the expression levels (Fig 5B, see also Appendix Figs S11 and S12).

We examined how expression levels varied depending on the codon identity at in-frame positions −2 and −1 (last two codons) (Fig 5C). Clearly, amino acid identity at both positions −1 and −2 had a strong influence on expression levels. Positively (K, R) and negatively (D, E) charged amino acids at either position led to a higher expression. In particular, the presence of one lysine residue at one of the two positions increased expression level by 1.48- to 1.55-fold compared to the average. Hydrophobic amino acids A, I, L, M, V, F, W, and Y at either position produced in general lower expression levels (for the two positions, mean normalized DAMRatio 0.86, standard deviation 0.08). Cysteine also led to lower expression levels (normalized DAMRatio 0.76–0.79). Unexpectedly, we did not observe any significant changes in expression in the presence of proline or threonine. The random library also contained the stop codons UAA and UAG at positions −2 and −1, which acted as internal controls. In the presence of a stop codon at position −1, the expression level was identical to the average (×1.01), recapitulating the effect of all possible C-terminal codons at position −2. When a stop codon was located at position −2, we observed an expression higher than the average (×1.23), due to the two lysine codons at positions −4 and −3 of the wild-type sequence.

We quantified the cooperativity (epistasis) between the amino acid identities at the two positions (Fig 5E) by computing the ratio $Q$ between the measured expression level of the pair and the expected expression level assuming that the effect of the two positions is independent. Most of the pairs showed almost no cooperative effect. Epistasis for the pairs KK-Stop, RK-Stop, and KR-Stop was negative, which most probably resulted from the saturation of the cumulative effect of C-terminal amino acids on protein expression levels. In fact, the presence of one lysine at either position −2 or −1 resulted in higher than average expression levels regardless of the identity of the other residue (Fig 5D). Negative epistasis was also found for the pairs [M,F,W,Y]-[A,V,I,L,M,F]-Stop (mean $Q$, 0.88, standard deviation 0.06), which led to the lowest expression levels (Fig 5D). The pairs PP-Stop and RP-Stop exhibited negative epistasis ($Q = 0.96$, norm. DAMRatio 0.86), while DP-Stop and EP-Stop showed positive epistasis ($Q = 1.10$, norm. DAMRatio 1.18). Apart from the lower expression for the PP dipeptide (norm. DAMRatio 0.81), these results disagreed with the depletion of DP and EP observed in the genomic analysis, and the fact that these two motifs were shown to induce high level of ssrA tagging in *E. coli* (Hayes *et al*, 2002). Overall, the expression levels varied up to 4.42-fold in the weak promoter library (3.51-fold in the strong promoter library), between the pair with the lowest expression, WL-Stop (MM-Stop), and the pair with the highest expression, KQ-Stop (NK-Stop).

The results were highly reproducible, as shown by the strong correlation of the normalized DAMRatio between the weak and the strong promoter libraries (Fig 5F), both at the level of the codon identity at position −1 (Pearson correlation 0.96) and at the level of amino acid pair (Pearson correlation 0.93). At the level of codon pairs, the lower number of reads available for each sequence led to higher variability (Pearson correlation 0.61, Appendix Fig S13, see also Appendix Figs S14 and S15). In the strong promoter library, the

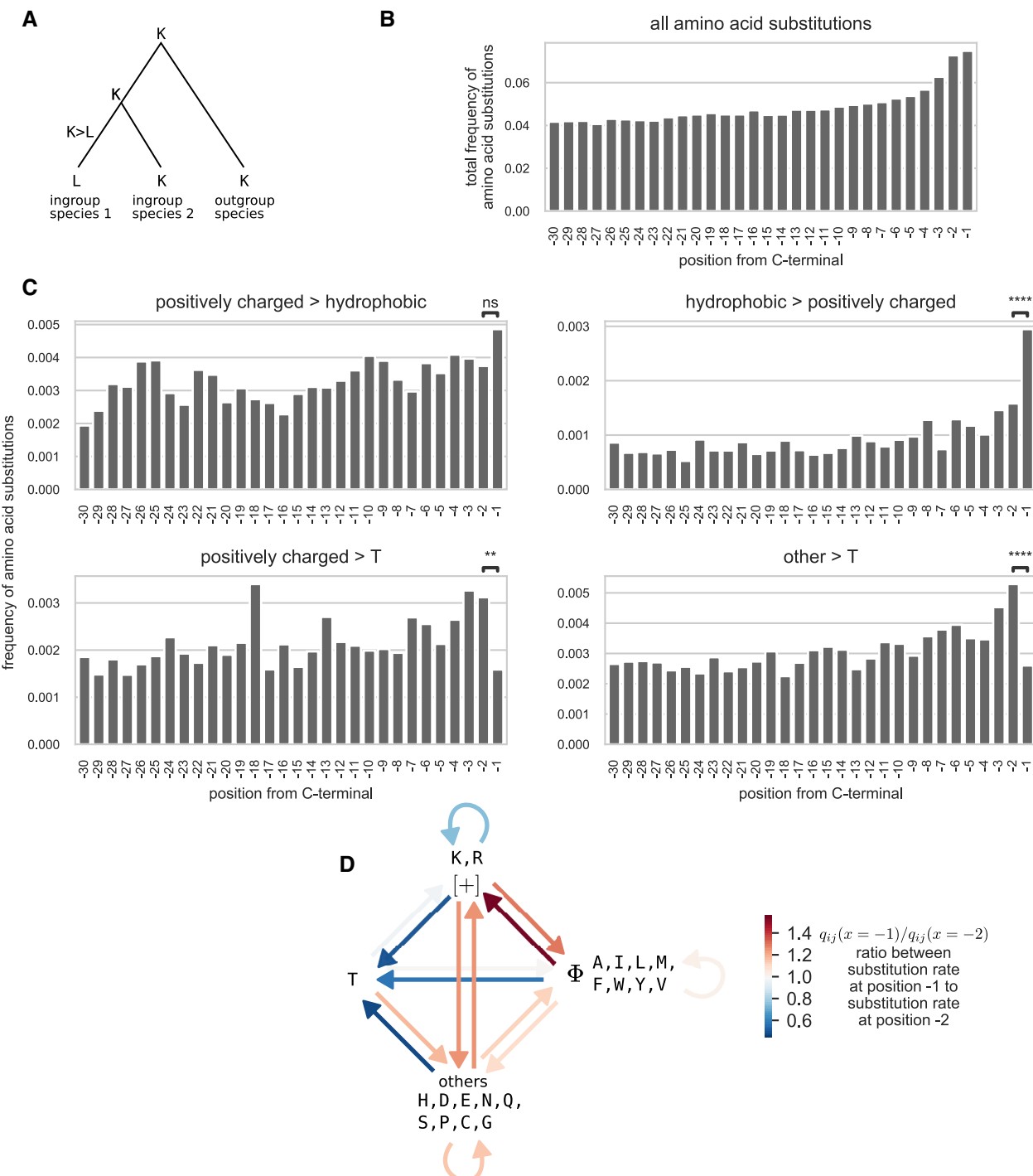

**Figure 4. Evolutionary analysis of amino acid substitution rates at the C-terminal.**

A  Amino acid substitutions were reconstructed in three closely related genomes, using the maximum parsimony principle. Only sites where either no changes were observed or one amino acid change was observed in one of the ingroup species were considered in the analysis.

B  Total frequency of all amino acid substitutions at each position from the C-terminal.

C  Substitution rates between amino acid groups at each position from the C-terminal. In order to allow reliable statistics, amino acids were grouped into the following categories: positively charged (K, R), hydrophobic (A, I, L, M, F, W, Y, V), threonine (T), and others (H, D, E, N, Q, S, P, C, G). Only 4 between-groups substitutions are plotted out of the 15 possible between-group and within-group substitutions. The same data were used as in Fig EV3. The difference of the substitution rates between positions −1 and −2 was tested by means of two-sided Fisher's test. Significance code: n.s. not significant for $P > 0.05$, ** for $P < 0.01$, **** for $P < 1e-4$.

D  Diagram of all possible between-groups and within-group substitutions. The color of the arrow denotes the relative change (ratio) between the substitution rate at position −1, $q_{ij}(x = -1)$, to the substitution rate at position −2, $q_{ij}(x = -2)$.

pattern of changes in expression was similar but less pronounced than in the weak promoter library, probably due to saturation effects.

We examined whether other sequence factors could play a role in determining the expression levels in the library. Secondary structure of mRNA is well-known to influence the rate of translation at

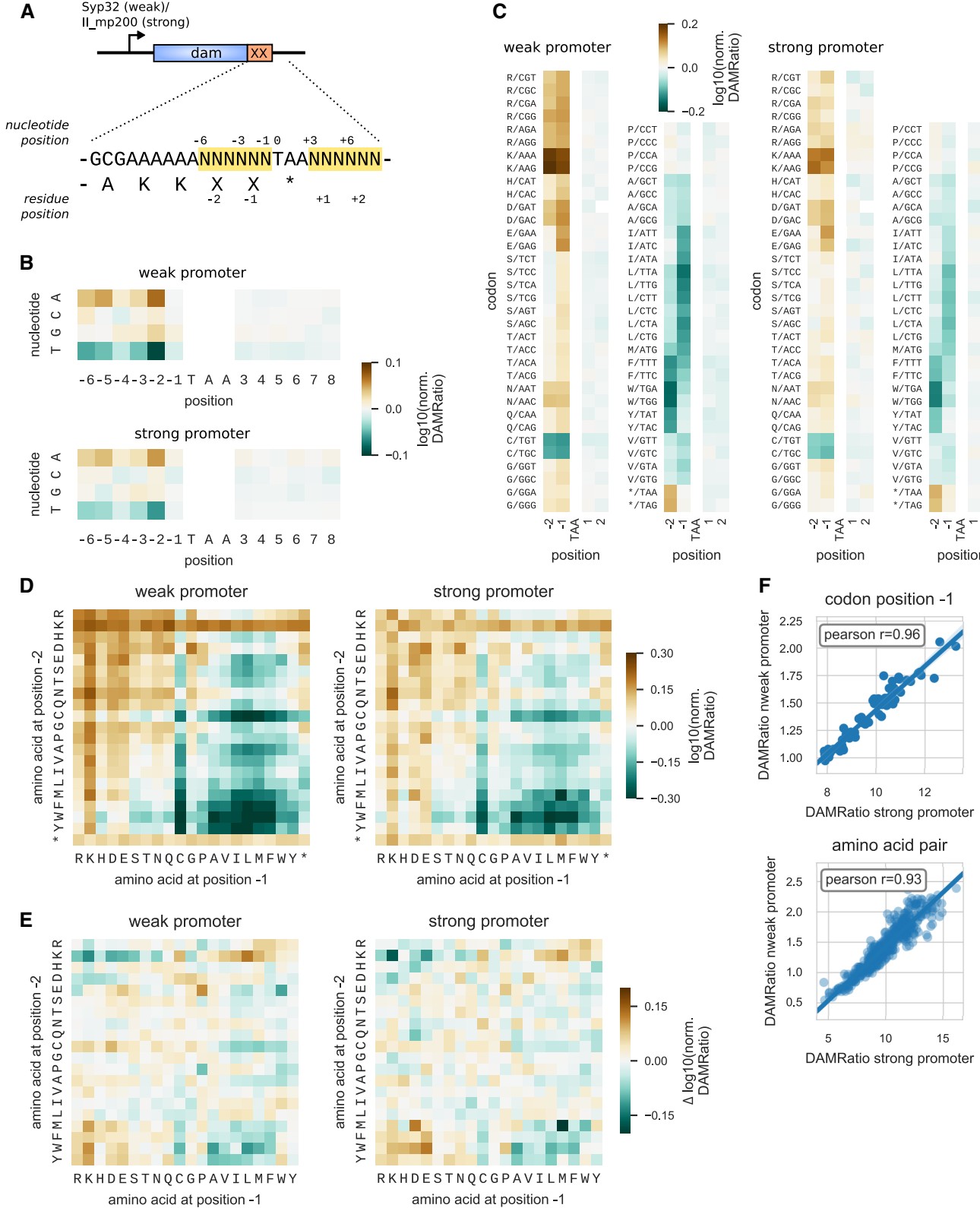

Figure 5.

◀

**Figure 5. Protein quantification of a randomized C-terminal library using ELM-seq.**

A A C-terminal library was created with the *dam* reporter gene, where the last 6 nucleotides before and the 6 nucleotides after the stop codon were randomized (indicated by Ns). Two versions of the library were created using either a weak or a strong promoter. Expression levels in *Mycoplasma pneumoniae* were measured by ELM-seq, a technique that combines DNA methylase as a reporter with methylation-sensitive restriction enzyme digestion and high-throughput sequencing. Protein expression level readout is reported as the $\log_{10}$ of the DAMratio relative to the average.

B Effect of nucleotide identity at each position on expression levels.

C Effect of codon identity at in-frame codon positions −2, −1 (upstream the stop codon), +1, and +2 (downstream the stop codon) on expression levels.

D Effect of C-terminal amino acid pair identity (last two codon positions before the stop codon) on expression levels.

E Cooperativity in the amino acid pair effect on expression levels (epistasis), as measured by the difference between expected and measured log10(DAMRatio).

F Correlation of expression levels between weak and strong promoter libraries, at the level of codon at position −1 and at the level of amino acid pair.

initiation (Cambray *et al*, 2018), elongation (Burkhardt *et al*, 2017), and termination (Del Campo *et al*, 2015). In our library, because we randomized 12 nucleotides around the stop codon, variations in the secondary structure could potentially affect translation termination efficiency. However, no correlation was found between predicted folding energy around the stop codon and expression levels (Appendix Fig S16).

In order to test whether the impact of the C-terminal sequence was specific to the Dam protein, we measured the expression levels of a different reporter gene (chloramphenicol acetyltransferase, *cat*), when adding one of the 20 amino acids to the C-terminal (Fig EV4). The results were in qualitative agreement (Appendix Fig S17, Pearson correlation 0.58 weak promoter, 0.61 strong promoter), and the same general trends were found: Hydrophobic amino acids led to lower expression than positively charged amino acids (Fig EV4 panel C, *t*-test independent, *P* = 0.015).

In conclusion, we showed experimentally that the identity of the last two amino acids impacts protein expression levels, with differences in protein abundance up to fourfold, with lysine leading to the highest expression levels, and hydrophobic amino acids leading to the lowest levels.

### C-terminal amino acid identity modulates protein degradation rate in *Mycoplasma pneumoniae*

Protein expression level results from the balance between synthesis and degradation rates, and C-terminal sequence composition could potentially influence one or both of these processes. In order to shed light on the underlying mechanism that drives the observed differences in protein abundance, we developed an experimental assay to measure protein degradation rates. In this assay, the firefly luciferase (*luc2*) reporter gene was fused to a C-terminal extension and placed under the control of a Tet-inducible system (Fig 6A). A representative set of amino acid extensions was chosen based on the results of the previous experimental assays and the biases found in the genomic study: NK, K, D, T, P, F, L, WL (in decreasing order of expression levels). We assumed that the C-terminal extension did not affect luciferase activity, such that the measured luminescence was proportional to the luciferase protein abundance.

We first examined the steady-state expression levels of the C-terminal luciferase variants (Fig 6B). As in the ELM-seq assay, variants that carried a charged residue at the C-terminal (D, K, NK) led to a higher expression than variants carrying a hydrophobic residue (F, L, WL) (independent *t*-test *P* = 2.7e-06). A C-terminal D extension resulted in the highest expression levels (2.04× compared to the expression level of the C-terminal P variant), while K and NK produced slightly lower expression levels (1.49× and 1.38×).

Hydrophobic residues (F, L) led to lower expression (0.80× and 0.70×), with the pair WL leading to the lowest expression level (0.40×). Overall, the difference in expression levels between the WL and the D variants reached 5.1-fold. Both C-terminal P and T variants yielded average expression levels (1.00× and 0.87×). Overall, these results were in agreement with the results of the ELM-seq assay (Pearson correlation coefficient 0.75, Appendix Fig S18).

In order to evaluate the impact of the C-terminal amino acid in protein decay, the luciferase gene expression was shut down for different time points (2, 4, 6, and 8 h) and the protein degradation rate of each C-terminal variant was derived by fitting the decrease in luminescence levels to an exponential decay (Fig 6C). We found significant differences between the C-terminal variants (one-way ANOVA test *P* = 2.2e-05), with variants carrying C-terminal hydrophobic residues (F, L, WL) showing a larger degradation rate than variants carrying C-terminal charged residues (D, K, NK) (independent t-test *P* = 3.6e-04). C-terminal L variant showed the fastest degradation rate (0.26 h$^{-1}$, half-life 2.67 h), while C-terminal D showed the slowest degradation rate (0.072 h$^{-1}$, half-life 9.67 h), with a difference of 3.6-fold.

We wondered how much of the variation in the steady-state expression level ($X^{s.s.}$) could be explained by the observed differences in degradation rates ($d$). By assuming a constant synthesis rate ($s$) of the luciferase, independent of the C-terminal variant, steady-state levels would result from the equilibrium between synthesis and degradation, i.e., $X^{s.s.} = s/d$. Linear regression of $X^{s.s.}$ to $1/d$ (Fig 6D) indeed showed that 85% of the variance of the steady-state expression levels could be explained by the variation in the degradation rate (ordinary least-squares regression, adjusted $R^2 = 0.85$). We emphasize that although steady-state expression levels and degradation rates were measured in the same experiment, they resulted from independent observations: While the former relied on the absolute luminescence and its normalization, the second only relied on the kinetics of the luminescence variation. Thus, the observed covariation is unlikely to be produced by a confounding common factor.

In conclusion, our results suggest that differential protein degradation rate could be one of the main mechanisms that drives the impact of C-terminal amino acid composition on protein expression levels.

## Discussion

Previous studies of biases in C-terminal composition considered a small number of bacterial species (Brown *et al*, 1990; Arkov *et al*, 1993; Björnsson *et al*, 1996; Rocha *et al*, 1999; Palenchar, 2008)

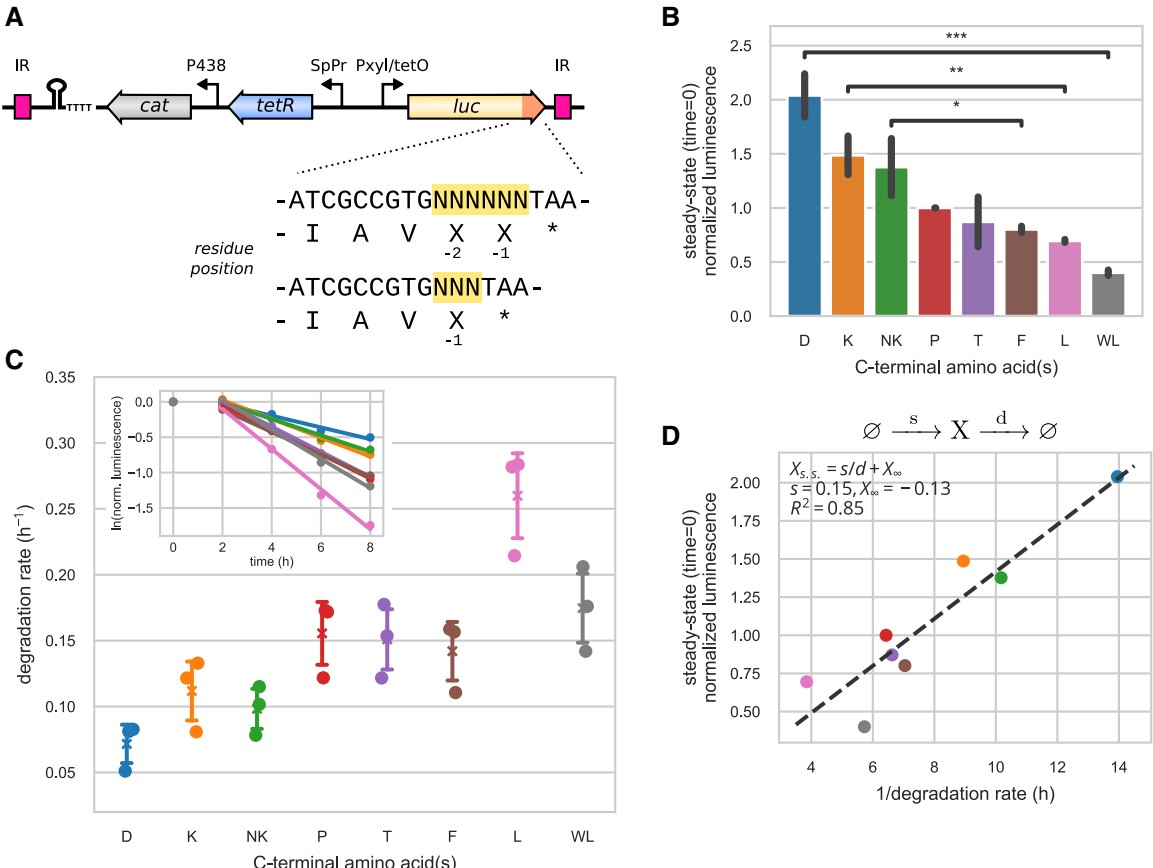

**Figure 6. Protein degradation assay of C-terminal variants.**

A   Scheme of the reporter system used to monitor protein degradation. The luciferase reporter gene (*luc*) is under the control of a Tet-inducible promoter and carries a C-terminal extension of one or two amino acids. The *tetR* repressor and *cat* resistance marker, which were also used for *luc* expression normalization, were cloned in the opposite direction to the *luc* gene. In addition, a terminator sequence shown as a hairpin structure was included to isolate the Tet-inducible promoter. IR denotes the inverted repeats of the mini-transposon vector.

B   Steady-state (time 0 h) luminescence levels of the C-terminal variants under fully induced conditions. Luminescence was first normalized by Cat expression, and then by the expression of the C-terminal P variant to show relative differences. Charged amino acids exhibited higher expression level than hydrophobic amino acids (independent *t*-test, NK > F, *P* = 3.8e-02, K > L, *P* = 3.3e-03, D > WL, *P* = 3.3e-04). Error bars show standard deviation of triplicate measurements. Significance code: * for *P* < 0.05, ** for *P* < 0.01, and *** for *P* < 0.001.

C   Gene expression was shut down at time 0 h by removing inducer from the medium. For each replicate and C-terminal variant, the time course of the luminescence normalized to time 0 h was fitted to an exponential decay (inset, example for one replicate) to derive degradation rates (main plot, cross, and whiskers denote the mean and standard deviation of triplicates).

D   Steady-state expression levels X result from the balance between synthesis, at rate *s*, and degradation, at rate *d*. Assuming a constant synthesis rate, variation in the degradation rate across C-terminal variants explained 85% of the variance in steady-state normalized luminescence X, as shown by linear regression of X with respect to 1/*d*.

which limited their level of resolution. In our study, we took advantage of the vast increase in the number of assembled bacterial genomes (Haft *et al*, 2018) to explore C-terminal composition biases across bacterial taxonomy. Here, we showed that the main pattern of composition biases at position −1, namely the enrichment for lysine and arginine, the depletion of threonine, and the depletion of hydrophobic amino acids, is conserved across all bacterial phyla. In fact, the biases for lysine and against threonine were found in the vast majority of the 280 taxonomic clades at the family rank (Appendix Fig S4). Similar biases were also found in eukaryotes, where positively charged residues are enriched at the C-terminal, while threonine and glycine are depleted (Berezovsky *et al*, 1999; Palenchar, 2008; Requião *et al*, 2017). Moreover, we

showed that this bias pattern was found in all protein functional categories. Thus, our results suggest that the main C-terminal amino acid composition biases are conserved across bacterial species.

We found further indications of a selective pressure acting specifically on the identity of the C-terminal residue. In particular, our results suggest that positive selection could favor positively charged amino acids, while purifying selection could disfavor threonine, at the C-terminal position −1. It could be interesting to study whether those selection forces are stronger in highly abundant proteins. A larger dataset of protein abundance data and a corresponding set of closely related protein sequences would be necessary to test such a relationship.

Both the enrichment for lysine and the depletion of threonine were stronger in highly abundant proteins, showing a clear association between protein abundance and C-terminal amino acid identity. In addition, hydrophobic amino acids V, I, L, and M were disfavored in all protein groups. In agreement with this, our expression assay revealed that C-terminal lysine, and to a lesser extent arginine, increased the expression levels of the *dam* reporter gene in *M. pneumoniae* and that the presence of hydrophobic amino acids decreased expression levels. Qualitatively similar results were obtained with two other reporter genes, supporting the notion that the observed modulations in expression are independent from the characteristics of the Dam protein. Altogether, our results suggest that the identity of the C-terminal amino acid modulates protein abundance in bacteria and is under selective pressure.

Protein abundance level results from the balance between synthesis and degradation rates, and C-terminal sequence composition could potentially influence one or both of these processes. In order to identify which underlying mechanism drives this phenomenon, we measured protein degradation rates of the luciferase reporter gene with short C-terminal extension variants. This assay revealed that differences in the degradation rate could explain a large part of the observed variation in luciferase steady-state expression levels. Moreover, the amplitude of this variation (3.6-fold) was comparable to the observed differences in Dam protein expression levels (4.42-fold and 3.51-fold in the weak and strong promoter library). These results suggest that differential protein degradation rate could be one of the main mechanisms that drive the impact of C-terminal amino acid identity on protein expression, at least in *M. pneumoniae*. In bacteria, apart from the universal N-terminal signals recognized by the N-end-rule pathway, several examples of degrons that target proteins to specific proteases have been identified (Sriram *et al*, 2011). For instance, the *E. coli* ClpXP protease recognizes the ssrA tag at the C-terminal of proteins (Gottesman *et al*, 1998; Sauer & Baker, 2011). The ssrA tag sequence varies across bacterial species, but exhibit a similar C-terminal LAA-coo⁻ motif, with the exception of Mollicutes, where the ssrA tag sequences are unusually longer and exhibit a NYAFA-coo⁻ conserved C-terminal motif (Gur & Sauer, 2008b). *Mycoplasma pneumoniae* does not possess ClpXP, but the ssrA tag is efficiently degraded *in vitro* by recombinant Lon protease (Gur & Sauer, 2008b). Interestingly, aspartic substitutions of the last two hydrophobic amino acids (FA-Stop) of the *M. pneumoniae* ssrA tag prevent the efficient degradation by recombinant Lon (Ge & Karzai, 2009). Despite the importance of these C-terminal residues, several other amino acids within the ssrA tag seem to be also involved for efficient recognition and degradation, suggesting the existence of multiple signaling elements. In our analyses, we found biases against hydrophobic amino acids at the last C-terminal position, which correlated with reduced levels of expression of three different gene reporters and increased degradation rates in *M. pneumoniae*. Interestingly, a similar observation was made for the halophilic archaeon *Haloferax volcanii* (Reuter *et al*, 2010), suggesting this could be a common mechanism to modulate protein levels. Protein degradation dependent on the universal Lon protease has been associated with degrons rich in aromatic and hydrophobic residues that would typically be buried in the hydrophobic core of native proteins (Gur & Sauer, 2008a). As C-terminal tails are most of the time exposed to the solvent (Jacob & Unger, 2007), the presence of a single hydrophobic residue could trigger partial recognition by Lon. We propose that further experiments to measure protein degradation rates of C-terminal reporter gene variants in protease-deficient bacterial strains could shed light on the detailed mechanism.

The results of our protein degradation assay also suggest that at least 15% of the variance in protein levels could be due to other mechanisms. Differential efficiency of translation termination is another possible mechanism that could affect protein synthesis rates. Proline at C-terminal can induce translation termination stalling in *E. coli* (Hayes *et al*, 2002), which is usually alleviated by the *ssrA* system. Regarding threonine, one of the few C-terminal sequence motifs identified that induce termination stalling in *E. coli* (Woolstenhulme *et al*, 2013) included threonine at position −1 with a hydrophobic-rich motif upstream (WILFXXT-Stop). Moreover, the threonine residue was shown to be essential for the stalling effect, suggesting that C-terminal threonine might reduce termination efficiency in specific contexts. However, despite threonine was found to be depleted in all bacterial species and in the Mycoplasma genus (odds ratio 0.41), and appeared to be under negative selection, the measured expression level for variants with C-terminal threonine was close to the average. The same happened with proline (odds ratio 0.25 in *M. pneumoniae*). Noteworthily, we observed a strong anticorrelation across all phyla between lysine and threonine biases, which could point to a common mechanism producing these two preferences. However, our experimental results showed that C-terminal threonine had no influence on protein expression level or degradation rate. While these assays were performed in only one species, they suggest that the bias against threonine observed in bacteria is due to a different unknown mechanism than the bias in favor of lysine.

Based on these observations, we postulate that the presence of threonine or proline at the C-terminal of a highly abundant protein could negatively impact the fitness of the bacterium without changing the protein expression level. For example, a moderate pausing at translation termination could increase the load of ribosomes on the mRNA while barely changing the total protein output, if termination is not the limiting rate in translation. The subsequent reduction in the pool of free ribosomes could decrease the expression of the other genes and lead to a fitness cost (Shah *et al*, 2013; Cambray *et al*, 2018). In this mechanism, the selection bias will be seen only at the −1 position and no gradient will be seen, which is the case for these amino acids.

The variability across taxonomy of the bias for proline also suggests that translation termination stalling at C-terminal proline could be species-specific. In this regard, an *in vitro* study of the release factor (RF)-mediated rates of peptide release has suggested that peptide release is particularly inefficient on peptides terminating with proline or glycine, for both RF1 and RF2, and especially in the absence of methylated RFs (Pierson *et al*, 2016). The addition of elongation factor P (EF-P) had no effect on the release rate suggesting that EF-P does not significantly promote peptide release on proline and glycine residues (Pierson *et al*, 2016). On the other hand, prokaryotic release factor 3 (RF3) plays a role both in quality control during elongation (Zaher & Green, 2011) and in termination, where it promotes the recycling of RF1/2 and increases the rate of translation termination (Matsumura *et al*, 1996; Freistroffer *et al*, 1997; Crawford *et al*, 1999; Baggett *et al*, 2017). While improving the efficiency and fidelity of termination, RF3 is not essential for growth in

*E. coli* (Mikuni *et al*, 1994) and is absent from many groups of bacteria (Margus *et al*, 2007). Interestingly, we found that depletion of proline at C-terminal was more pronounced in taxonomic clades lacking RF3, like *M. pneumoniae* (*t*-test, *P* = 5.14e-4) (Appendix Fig S19). This suggests that translation stalling at C-terminal proline might be dependent on the absence of RF3, which would explain the variability in proline biases across taxonomy.

In conclusion, our study reveals an important impact of the C-terminal residues in protein expression. We propose that the preferences for C-terminal amino acid composition could be easily taken into account in the optimization of heterologous protein expression. Our results suggest that the addition or substitution of one or two lysine residues at the C-terminus of a protein sequence could increase expression, in particular when the original protein sequence contains hydrophobic amino acids at the C-terminus. Moreover, more subtle modulations in the protein expression levels, often needed in the design of synthetic circuits, could be achieved by controlling the identity of the C-terminal residues.

# Materials and Methods

### Reagents and Tools table

| Reagent/Resource | Reference or Source | Identifier or Catalog Number |
|---|---|---|
| **Experimental models** | | |
| *Mycoplasma pneumoniae M129* | Richard Herrmann lab | |
| *Escherichia coli* TOP10 | Invitrogen | C404003 |
| *Escherichia coli* DH5α | NEB | C2987H |
| *Escherichia coli* MegaX DH10B T1 | Invitrogen | C640003 |
| **Recombinant DNA** | | |
| Plasmids | This study | Table EV3 |
| **Oligonucleotides and sequence-based reagents** | | |
| *Primers* | This study | Table EV2 |
| **Chemicals, enzymes and other reagents** | | |
| CAT ELISA Kit assay | Roche | 11363727001 |
| BCA Protein assay Kit | Pierce | 23225 |
| ONE-Glo Luciferase Assay System | Promega | E6110 |
| EcoRV | NEB | R0195S |
| XhoI | NEB | R0146S |
| BamHI | NEB | R0136S |
| EcoNI | NEB | R0521S |
| NotI | NEB | R0189S |
| Eco147I | ThermoScientific | FD0424 |
| NsiI | NEB | R0127S |
| T4 ligase | NEB | M0202S |
| Phusion High-Fidelity DNA Polymerase | ThermoScientific | F530S |
| Chlorampehnicol | Sigma | C0378 |
| Tetracycline | Sigma | T7660 |
| Gentamycin | Sigma | G1397 |
| HEPES | Sigma | H4034 |
| Sucrose | Sigma | 84097 |
| **Software** | | |
| Python v3.6.9 | https://www.python.org | |
| CD-HIT v4.6 | Li and Godzik (2006); Fu *et al*, (2012) | |
| ETE 3 (python package) | Huerta-Cepas *et al* (2016) | |

**Reagents and Tools table** (continued)

| Reagent/Resource | Reference or Source | Identifier or Catalog Number |
|---|---|---|
| Statsmodels v0.9.0 (python package) | Seabold and Perktold (2010) | |
| Scipy.stats v1.1.0 (python package) | Virtanen *et al* (2020) | |
| Eggnog-mapper v0.12.7 | Huerta-Cepas *et al* (2017) | |
| NCBI RefSeq (database) | O'Leary *et al* (2016); Haft *et al*, (2018) | |
| PSORTb v3.00 (database) | Peabody *et al* (2016) | |
| paxDB release 4.1 (database) | Wang *et al* (2015) | |
| Alignable Tight Genomic Clusters (ATGC) (database) | Kristensen *et al* (2017) | |
| PAML package v4.9j | Yang (2007) | |
| Vienna RNA package v2.0 | Lorenz *et al* (2011) | |
| GNU parallel | Tange (2011) | |
| **Other** | | |
| Gene Pulser XCell™ electroporation system | Bio-Rad | |
| Infinite M200 plate reader | Tecan | |
| HiSeq 2500 sequencing platform | Illumina | |

## Methods and Protocols

### Bioinformatics analysis

#### Genomic and protein sequences

The main dataset in our analysis is the full set of annotated prokaryotic genomes of the NCBI RefSeq database (O'Leary *et al*, 2016; Haft *et al*, 2018). We selected the RefSeq collection for several reasons: (i) consistency and high quality in the gene annotation using a single annotation pipeline, (ii) classification of species using the NCBI taxonomy, and (iii) set of reference and representative genomes with curated annotation and balanced taxonomic diversity. We first downloaded the assembly summary file that reports metadata for the genome assemblies on the NCBI genomes FTP site at ftp://ftp.ncbi.nlm.nih.gov/genomes/refseq/ assembly_summary_refseq.txt, on February 9, 2017. We selected the genomes in the "representative" or "reference" genome sets and with a "complete" genome assembly status (no contigs). All 1,582 selected genome files were downloaded in GenBank format from the ftp website following ftp paths given in the summary file. For each genome, protein sequences together with codon sequences, stop codons, and codon tables were extracted from the genome annotations.

#### Clustering

Protein families with highly similar sequences are very common in bacteria and are often the result of gene duplication. In our analysis, we seek to avoid over-representation of duplicated proteins in the same bacterial species. In particular, duplicated proteins with identical C-terminal regions could bias the analysis of C-terminal amino acid composition. We applied a two-steps clustering to protein sequences in each bacterial species, to remove protein sequences that present both a high overall sequence identity and a high identity at their C-terminal region. We ran the CD-HIT clustering method (version 4.6) (Li & Godzik, 2006; Fu *et al*, 2012) with 80% identity threshold of the overall protein sequences. The resulting clusters composed of more than one sequence were clustered again by

running CD-HIT with 85% identity threshold on their C-terminal regions (20 last amino acid positions). We kept only one representative sequence for each of the final clusters. For each bacterial species, we thus obtain the set of non-redundant sequences. In addition, we also filter out proteins with length smaller than 50 residues. In total, we obtained 4,897,860 non-redundant sequences out of 4,934,952 (0.75% sequences removed).

#### Taxonomy

We used the ETE's ncbi_taxonomy python module (Huerta-Cepas *et al*, 2016) which provides utilities to efficiently query the NCBI Taxonomy database (Federhen, 2012). The NCBI Taxonomy database was downloaded as of February 2, 2017. For each genome in the assembly summary, the organism taxid was searched for in the taxonomic tree and taxonomic lineage information was retrieved. After removing non-bacterial genomes, we obtained 1,582 genomes in the bacterial superkingdom, summing up 4,934,952 sequences.

Genomes were grouped by taxonomy at different ranks, ranging from superkingdom to species. In total, we obtained 2,859 taxonomic groups which contained at least one genome. Our pipeline for C-terminal analysis was applied to the protein sequence sets of each of the taxonomic groups, by grouping all non-redundant protein sequences in the taxon. We used the ETE package (Huerta-Cepas *et al*, 2016) to draw the taxonomic tree and the C-terminal biases of each clade.

#### Phylogenetic tree

We approximated evolutionary distances between the taxonomic phyla based on the bacterial phylogenetic tree of Lang *et al* (2013), which used 761 bacterial taxa, inferred from a concatenated, partitioned alignment of 24 genes using RAxML. We mapped bacterial species to the NCBI taxonomy, obtaining 263 species with assigned phylum. While a phylogenetic tree will never be fully consistent with the taxonomy tree (Parks *et al*, 2018), we aimed at obtaining approximate evolutionary relationships at a high level of the taxonomy. Thus, we reconciled the NCBI phyla groups with the

phylogenetic tree by pruning together monophyletic groups of species with the same phylum, discarded 38 species that were a minority in near-monophyletic groups. As recently described (Parks et al, 2018), the Tenericutes phylum should be included in the Firmicutes as a class. We relocated the Tenericutes phylum as a close sibling to Firmicutes. We obtained an approximated phylogenetic tree at the level of phyla.

*Analysis of C-terminal bias*
Amino acid composition of the C-terminal and N-terminal regions was analyzed as follows. Protein sequences were split into three parts: N-terminal region (20 first amino acids), C-terminal region (20 last amino acids), and bulk (remaining part of the sequence). The N-terminal region was discarded in order to remove well-known N-terminal sequence biases from the bulk composition. Position-dependent amino acid counts for the terminal regions were compared to bulk amino acid counts, and two-tailed Fisher's exact test [scipy.stats package version 1.1.0 (Virtanen et al, 2020)] was used to test for the significance of enrichment or depletion of amino acid "X" at position $j$ in the C-terminal region, where $j \in [-20, \ldots, -1]$. The resulting odds ratios and $P$-values were computed for all 20 amino acids and 20 positions. Multiple testing correction was applied using the Benjamini–Hochberg method [from the statsmodels python package v0.9.0 (Seabold & Perktold, 2010)], with family-wise false discovery rate of 5%. Only significant biases after multiple testing correction were plotted. We followed the same method to study the bias in codon composition and for the composition bias in amino acid pairs and codon pairs (hexamers) at the last two positions.

In the cases where the count of proteins was very low, even moderately strong biases would not appear as significant. Those cases of underpowered statistics were indicated in the plots with a hashed pattern. More precisely, based on the count of amino acid "X" in the bulk and its frequency, we computed the minimal count of expected observations of amino acid "X" at the C-terminal such that an odds ratio of 0.5 between observed and expected counts would lead to a $P$ value lower than 0.05. For example, low frequency amino acids such as cysteine often had too low counts to be able to detect biases in the frequency at C-terminal.

In order to study the interaction between biases at two positions, or epistasis effect, we compared the frequency of an amino acid pair at the last two positions to its expected frequency under the assumption that both positions are independent (null model). More precisely, we defined the expected frequency as the product of the frequency of amino acid at position $-2$ and the frequency of amino acid at position $-1$. The deviation of the observed frequency from the expected was tested with the binomial test (scipy.stats.binom_test version 1.1.0).

*Disordered regions*
Prediction of disordered regions for 1,305 bacterial genomes was downloaded from the $D^2P^2$ database (Oates et al, 2013) on July 31, 2019. Regions that were predicted as disordered by all six predictors (consensus of 100%) were considered.

*Stop codon context*
Protein sequences were classified by their stop codon context. Then, codon composition bias was analyzed as above within each class of stop codon context independently and multiple testing correction was applied within each class.

We also analyzed the relationship between the codon biases at position $-1$ and the identity of the codon third nucleotide in each stop codon context, by comparing the distribution of odds ratios of all codons for the selected 14 phyla (Appendix Fig S8). We emphasize that the differences in biases that we found could not be explained by global variations in the $3^{rd}$ nucleotide frequency, which could be influenced by the codon usage bias and genomic GC content of the bacterial species (Sharp et al, 2010; Plotkin & Kudla, 2011), because each individual codon frequency at position $-1$ was compared to its frequency in the bulk.

*Membrane proteins*
We downloaded from the database PSORTb version 3.00 (Peabody et al, 2016), which contains predicted subcellular localization for bacterial and archeal genomes, the full database tables for gram-positive and gram-negative bacteria. Based on RefSeq accession numbers, localization information could be assigned to 1,442,202 protein sequences out of 5,125,116 in our database. We then further selected 364 species in our database in which at least 80% of the proteins had localization information. Protein localized to the cytoplasmic membrane or to the outer membrane was grouped together as membrane proteins.

*COG categories*
In order to assign the Clusters of Orthologous Groups (COGs), functional category to every protein in our NCBI-based database, eggnog-mapper version 0.12.7 (Huerta-Cepas et al, 2017), was run in the hidden Markov model (HMM) search mode against the bactNOG database. This mapper assigns functional annotation based on fast orthology assignments using precomputed clusters and phylogenies from the eggNOG database. The COG functional category is inferred from best matching orthologous group. The mapper failed to compute the assignment for 42 species. We could assign orthologous group to 4,213,953 protein sequences out of the total of 4,871,983 (86%).

*Protein abundance*
Protein abundance datasets were downloaded from the paxDB database release 4.1 (Wang et al, 2015). We selected the 24 bacterial species that were also present in our RefSeq-based database based on NCBI taxa id. Protein sequences were matched in two steps, first by matching paxDB ids to UniProt ids using the paxDB provided mapping file, then by matching UniProt ids to RefSeq ids using the UniProt mapping file. We obtained 62,093 RefSeq proteins (out of 78,483 in total) with at least one paxDB protein match. For the RefSeq proteins with multiple paxDB protein matches (1,515 cases), the average protein abundance was computed. The resulting protein abundance coverage varied greatly among species. From the 24 species, only those where at least 40% of RefSeq protein sequences were assigned to an abundance value were kept (13 species, 40,002 RefSeq sequences, 26,935 with abundance value). Protein abundance values from the paxDB were given in relative values of part per million (ppm), which we assumed could be directly compared between species.

We then categorized proteins into three unequal bins: low abundance (0–20 percentiles), medium abundance (20–80 percentiles),

and high abundance (80–100 percentiles). We applied the same analysis of bias as described above independently to the three groups of proteins. Note that the resulting biases at the C-terminal are relative to the amino acid bulk frequencies of each group of proteins, which may vary.

*Evolutionary analysis*

Genomic data were retrieved from the updated version of the Alignable Tight Genomic Clusters (ATGCs) database (Kristensen *et al*, 2017). In order to reconstruct mutations in protein-coding DNA by the parsimony principle, we used triplets of closely related species. The triplets of species were chosen by following the same approach as in Rogozin *et al* (2016). We first selected a pair of species with *dS* in the range 0.2–1.0, preferentially choosing those as close to 0.2 as possible, to balance the requirements for a sufficient number of substitutions for reliable analysis and for the lack of mutational saturation. Then, the third species was chosen such that the distance from each member of the initially selected pair of species was at least 1.2 (preferably 1.5, but not more than 2) times greater than the distance within the pair, so that it would represent an unambiguous outgroup. We used the precomputed *dS* distances between any two members of an ATGC, defined as the genomic median value among all pairs of genes. A total of 57 species triplets were selected. Precomputed alignments of sequences in each ATGC-COG (Cluster of Orthologous Genes) were retrieved from the ATGC database. In order to simplify the evolutionary analysis, a single "index ortholog" was chosen for each species, defined as the most well-conserved member of an orthologous family from each of the included genomes.

One difficulty when studying the conservation of the C-terminal region is the prevalence of evolutionary events that lead to an extension or shortening of the 3′ part of the protein-coding sequence (Vakhrusheva *et al*, 2011). Most of these events are caused by a point mutation either in the stop codon, in which case the protein sequence extends up to the next downstream in-frame stop codon in the 3′UTR, or by a point mutation in the coding sequence that gives rise to new stop codons, usually close to the C-terminal, in which case the protein sequence is shortened. Other possible events include indels in coding regions close to the C-terminal that lead to frameshifts and use of out-of-frame stop codons. In order to simplify our analysis, we excluded variable in length C-terminal regions and selected orthologous groups for which the multiple sequence alignment (MSA) of protein sequences of the triplet presented no gaps for the last 3 columns (135,173 out of 396,240 groups in total, or 34%).

We used the maximum parsimony approach to infer ancestral nucleotide sequences [pamp tool in PAML package version 4.9j (Yang, 2007)], using the precomputed phylogenetic tree of the ATGC genome group. Despite the limitations of the maximum parsimony approach, it has been shown to produce accurate ancestral states when the evolutionary distances between the protein sequences are small (Kolaczkowski & Thornton, 2004; Matsumoto *et al*, 2015), which is the case for the selected species triplets.

Substitution rate between amino acid $X_1$ to amino acid $X_2$ was computed by counting the number of substitution events and dividing by the total number of sites with amino acid $X_1$ for the ancestral state. Only cases where the outgroup species amino acid was identical to that of one of the ingroup species were considered in the analysis.

In order to allow reliable statistics, amino acids were grouped into the following categories: positively charge (K, R), hydrophobic (A, I, L, M, F, W, Y, V), threonine (T), and others (H, D, E, N, Q, S, P, C, G). The median of between-groups substitution counts at the position −1 was 263, the minimum 15 ($T >$ positively charged) and the maximum 2,049 (other > other).

We estimated the average evolutionary rate of each gene by computing the dN/dS ratio, using maximum-likelihood estimates calculated using the CODML program of the PAML v4.9j package.

*mRNA secondary structure prediction*

Folding energy was computed using the ViennaRNA Package (Lorenz *et al*, 2011) with default parameters in a window ranging from 30 nucleotides upstream to 37 nucleotides downstream the randomized C-terminal sequence, both for the weak and the strong promoters.

**Experimental assays**

**Construction of C-terminal *cat* derivatives and CAT expression quantification**

Mini-transposon plasmids containing *cat* derivatives (Table EV3), in which one of the 20 amino acids was added to the C-terminal end, were constructed as follows. The *cat* reporter gene was first amplified with primer cat_mp200Pr_F and the corresponding reverse primer listed in Reagents and Tools Table (Table EV2). Each reverse primer incorporates at its 5′ end, the EcoRV restriction site and the codon sequence of one of the 20 amino acids. The corresponding PCR products were then used as templates for a second PCR, in which the cat_mp200Pr_F2 forward primer was used instead. This primer contains at its 5′ end the XhoI restriction site plus the mp200 promoter sequence to drive *cat* expression. Finally, all PCR products were cloned into a *XhoI*/*EcoRV*-digested pMTn*TetM438* mini-transposon vector (Pich *et al*, 2006).

To obtain *M. pneumoniae* M129 strains expressing the different C-terminal *cat* derivatives, all twenty constructs were transformed separately by electroporation as previously described (Pich *et al*, 2006) with few modifications. Briefly, 80 μl of cells (with approximately $10^9$ cells/ml) resuspended in electroporation buffer (8 mM HEPES·Na pH 7.2, 272 mM sucrose) was mixed with 2 μg of DNA in a 1-mm gapped electroporation cuvette (Bio-Rad). Cells were incubated on ice for 15 min and electroporated using the Gene Pulser XCell™ electroporation system (Bio-Rad) with the pulse controller set at 1.25 kV, 25 μF, and 100 Ω. After 15 min of incubation on ice, cells were resuspended in 900 μl of Hayflick medium, incubated 2 h at 37°C, and the pool of transformants selected in Hayflick medium containing 2 μg/ml of tetracycline.

For quantification of CAT expression, we used an ELISA-based assay (CAT ELISA Kit assay, Roche). Briefly, *M. pneumoniae* transformant pools were grown at 37°C in 25-cm² flasks containing 5 ml of Hayflick medium supplemented with 2 μg/ml of tetracycline. After 72 h, cells were washed three times in PBS and lysed with the CAT ELISA Kit lysis buffer and CAT expression determined following the manufacturer's instructions. Approximately, 25 ng of cell lysate was used per ELISA plate well and absorbance recorded using a Tecan Infinite M200 plate reader. CAT expression was normalized by total protein amount and determined by the bicinchoninic acid (BCA) reagent kit (Pierce).

### Protein degradation assays

To determine the influence of the last C-terminal amino acid in protein degradation, we developed a reporter system based on the firefly luciferase (*luc2*) gene. In this system, the expression of the luciferase gene is isolated by a terminator sequence and controlled by a Tet-inducible system (Mariscal *et al*, 2016). As shown in Fig 6, the reporter system also contains the *cat* gene, used to select transformants and to normalize *luc2* expression.

To construct this reporter system, a PCR fragment enclosing the terminator sequence, the *cat* gene, and the inducible platform containing the *tetR* repressor plus the inducible promoter was amplified with primers p_IndPlat_F and p_IndPlat_R (Reagents and Tools Table, Table EV2) using an in-house plasmid as template. Additionally, luciferase derivatives ending with K, D, L, F, P, T, WL, or NK amino acids were amplified using primers Ind_luc_F and the corresponding reverse primer listed in the Reagents and Tools Table (Table EV2). Each reverse primer incorporates the codon sequence of the C-terminal amino acid tested. Finally, the inducible platform and the corresponding *luc2* derivative were cloned by Gibson assembly into a *Xho*I/*Bam*HI-digested pMTnCat mini-transposon vector (Burgos & Totten, 2014) using the Gibson tails added in the primers. Finally, the mini-transposon vectors containing the different reporter systems (Table EV3) were transformed separately by electroporation in *M. pneumoniae* M129 strain as described above, and the resulting transformants were selected in Hayflick medium containing 20 μg/ml of chloramphenicol.

A protein degradation assay to monitor luciferase decay was performed as follows. *Mycoplasma pneumoniae* strains expressing the different reporter systems were cultured overnight in wells of 96-well plates under inducing conditions (Hayflick medium supplemented with 20 μg/ml of chloramphenicol and 100 ng/ml tetracycline). Then, luciferase gene expression was shut down for different time points (0, 2, 4, 6, and 8 h) by removing the inducer from the medium, and adding fresh medium without inducer. Luminescence from luciferase activity was detected for each time point by using a Tecan Infinite M200 plate reader and the ONE-Glo™ Luciferase Assay System (Promega) following the manufacturer's instructions. CAT expression measured by ELISA as described above (CAT ELISA Kit assay, Roche) was also used to normalize luciferase expression across the different derivatives. All measurements were performed from three biological replicates.

The degradation rates were derived by fitting the time course of the normalized luminescence to an exponential decay. The luminescence values were first normalized by the luminescence at time 0, for each variant and replicate. Then, the decay rates were derived by linear regression (ordinary least squares (OLS) of the statsmodel v0.9.0 python package) to the natural logarithm of the normalized luminescence values for time points 2, 4, 6, and 8 h (Appendix Fig S20).

### ELM-seq

We followed the same protocol as described in Yus *et al* (2017). Briefly, DNA adenine methylase (Dam) methylates multiple GATC sites that are located downstream of the *dam* coding sequence, in close proximity to the C-terminal randomized sequence. Dam expression level modulates the probability of having a site methylated, which is "measured" by differential digestion using

methylation-sensitive restriction enzymes that cut the GATC sequence. Ultra-sequencing of the region of interest allows to determine both sequence identity and expression level as reported by the ratio of reads between the two digested samples (DAMRatio).

### Dam cloning and library preparation

A version of the mini-transposon plasmid pMT85 (Table EV3) was prepared by linker ligation, in order to accommodate the "reporter cassette" (i.e., the 4xGATC sites) at the C-terminus, followed by an endogenous (MPN517) terminator (so that all RNAs had a similar length). Briefly, two linker pairs, F_sp_term and R_sp_term, and F_term_NI and R_term_NI (Table EV2), were annealed in two independent reactions and ligated to pMT85 that had been open-cut with NsiI and EcoNI. In the next step, the library was generated. The Gibson assembly reaction consisted in three pieces: the vector, cut-opened with NotI and Eco147I (introduced in a linker in the previous step), a PCR of dam, carrying the promoter in the forward oligos (F_P8_dam or F_mp200_dam and R_Cter_dam), and a linker with a C-terminus random region that was extended with Klenow in order to get the complementary random strand (F_damN_sp and R_damN_sp, Table EV2). The random region was ordered to have 40% GC content, similar to that of *M. pneumoniae* genome.

### ELM-sequencing

Procedure for DNA-seq was as previously described (Yus *et al*, 2017), except for some details (Appendix Fig S21). Contrary to the previous protocol, paired-end sequencing was chosen. In the final PCR step, a mix of equimolar forward oligos with a random region of increasing length (from 0 to 5 random nucleotides, see oligos in Table EV2: F_PE_Cdam, F_PE_n_Cdam, F_PE_n2_Cdam, F_PE_n3_Cdam, F_PE_n4_Cdam, F_PE_n5_Cdam), was designed in order to introduce enough diversity in the first Illumina DNA sequencing cycles. Two reverse oligos were used (R_PE_i6_Cdam and R_PE_i12_Cdam, Table EV2) alternatively, to be able to perform multiplexing. DpnI-treated DNA was amplified during 12 cycles, whereas MboI-treated DNA had to be amplified during 15 cycles.

### DAMratio analysis

We followed the analysis pipeline described in Yus *et al* (2017). Raw reads were filtered for both the common C-terminal Dam coding sequence and the varying downstream sequence according to the specific study as following CGCGAAAAAANNNNNNTAANN NNNNCAGGCCTTGA. In order to reduce the technical variability in the DAMRatio, we filtered out the sequences that did not have at least 30 reads in either one of the two enzyme-treated samples. In the end, 161,227 and 194,856 different sequence variants remained for strong and weak promoters, respectively. Higher thresholds resulted in the same or lower correlation between codon pair DAMRatios of the weak and strong promoter libraries.

Sequences that contained the GATC motif may introduce a bias in the measurement of the DAMratio and thus were filtered out prior to the analysis. Indeed, the number of GATC sites can modify the distribution of the DAMratios for a given library design (Yus *et al*, 2017). In the case of the weak promoter, sequences containing GATC showed a lower DAMratio than the other sequences. In the case of the strong promoter, the opposite trend was observed.

### *Epistasis*

In order to compute the intensity of the cooperative effect between the identities of the last two amino acids on the expression level, we first assumed that the effect of each amino acid is independent, such that the expected expression level of an amino acid pair would be

$$R_{ind}(aa_{-2}, aa_{-1}) = R(aa_{-2})R(aa_{-1})$$

where $R(aa_i)$ is the relative expression level of all sequences with amino acid $aa$ at position $i$, as measured by the DAMRatio normalized to the average. The cooperativity effect (or epistasis) was then computed as the ratio of the measured expression level of the pair to the expected one,

$$Q(aa_{-2}, aa_{-1}) = \frac{R(aa_{-2}, aa_{-1})}{R_{ind}(aa_{-2}, aa_{-1})}$$

which in log scale reads,

$$\log 10(Q(aa_{-2}, aa_{-1})) = \log 10(R(aa_{-2}, aa_{-1})) - \log 10(R(aa_{-2})) - \log 10(R(aa_{-1}))$$

## Data and software availability

All the bioinformatics analyses were done using custom Python 3.6 scripts. Some of the Bash scripts made use of GNU parallel (Tange, 2011). Source code is available at GitHub repository https://github.com/webermarcolivier/c-terminal. The datasets produced in this study are available in the following databases: ELM-seq data: ArrayExpress E-MTAB-8223 (https://www.ebi.ac.uk/arrayexpress/experiments/E-MTAB-8223).

**Expanded View** for this article is available online.

### Acknowledgements

We acknowledge the support of the Spanish Ministry of Science and Innovation to the EMBL partnership, the Centro de Excelencia Severo Ochoa, and the CERCA Programme/Generalitat de Catalunya. This project has received funding from the European Union's Horizon 2020 research and innovation program under grant agreement no. 634942 (MycoSynVac) and 670216 (MYCOCHASSIS). ML-S acknowledges the support from FEDER project from Instituto Carlos III (ISCIII, Acción Estratégica en Salud 2016) (reference CP16/00094). The authors would like to thank the CRG Genomics Unit for assistance with the design and sequencing of the ELM-seq library.

### Author contributions

MW, RB, EY, J-SY, ML-S, and LS conceived and designed the study. MW conceived and performed all bioinformatics analyses. EY designed the library and performed the ELM-seq experiment. MW and J-SY analyzed the ELM-seq results. RB designed and performed the CAT and luciferase reporter assays. LS and ML-S provided direct supervision. MW, RB, J-SY, ML-S, and LS wrote the manuscript. All authors read and approved the final manuscript.

### Conflict of interest

The authors declare that they have no conflict of interest.

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
