## [Review Process File · Molecular Systems Biology]

Impact of C-terminal amino acid composition on protein expression in bacteria

Marc Weber, Raul Burgos, Eva Yus, Jae-Seong Yang, Maria Lluch-Senar, and Luis Serrano

Review timeline:

Submission date:	28 th August 2019
Editorial Decision:	10 th October 2019
Revision received:	8 th March 2020
Editorial Decision:	26 th March 2020
Revision received:	7 th April 2020
Accepted:	9 th April 2020

Editor: Maria Polychronidou

Transaction Report:

1st Editorial Decision

10th October 2019

Thank you again for submitting your work to Molecular Systems Biology. We have now heard back from the three referees who agreed to evaluate your study. As you will see below, the reviewers acknowledge that the study seems potentially interesting. They raise however a series of concerns, which we would ask you to address in a major revision.

Without repeating all the comments listed below, some of the more fundamental issues refer to the need to:

- include further analyses to investigate the mechanisms underlying the effect of the C-terminus composition on expression levels. We understand that such analyses are likely to be time consuming, and we do not think that a detailed dissection of the mechanisms is mandatory. However, we do agree with reviewer #2 that offering some level of mechanistic understanding would significantly enhance the impact of the study and we would therefore encourage you to include analyses in this direction.

- perform additional experimental analyses and controls to strengthen the main conclusions. Both reviewers #2 and #3 provide constructive suggestions in this regard.

All other issues raised by the reviewers would need to be convincingly addressed. Please feel free to contact me in case you would like to discuss in further detail any of the issues raised.

REFeree REPORTS

Reviewer #1:

The manuscript by Weber et al revisits the question of how variation at the protein C-terminal affects gene expression - this time focusing on bacteria and making use of the huge increase in data in the last decade thanks to genome sequencing efforts. Through a series of comparative genomics analysis methods, it is clearly shown that some amino acids are more common or rare at the gene C-terminus than random chance would allow. The analysis reveals that this is less likely to be due to codons or RNA structure and is more likely to be due to the amino acid itself. The general trends of which amino acids are enriched and depleted are seen across all bacteria (and seemingly even eukaryotes as briefly mentioned in the discussion) so it seems to be a largely universal rule. Interestingly some amino acid enrichment/depletion trends are only seen in some parts of the bacterial taxonomy, whereas others are universal. This suggests two mechanisms are going on - one universal and one more species-specific. Furthermore, the role of the protein doesn't seem to play much of a role in whether these rules are important or not.

The work also looks at how the final amino acids in proteins affect gene expression, first by their bioinformatics methods, and then using an innovative assay to assess a synthetic expression library in a *Mycoplasma*. This is a cool addition to an otherwise quite dry comparative genomics bioinformatics analysis paper. It provides good evidence that the amino acids chosen link to increased or decreased expression level in the cases seen in the above analysis.

Overall I'm broadly supportive of publication of this high-quality and interesting research. It focuses on work that has very broad relevance to anyone interested in bacterial gene expression (probably half of this journal's readership!). It lacks a major wow factor in terms of not ever revealing the actual mechanisms that are driving the selection of amino acid choice at the C-terminus, but that is not really needed to be honest.

I recommend publication in MSB following minor corrections as suggested below.

1. The figure panel order and supplementary figure order is not in line with the main text - e.g. Fig 1D is talked about before Fig 1B. This was horribly confusing and made reading the paper very annoying. Please make sure that panels and supplementary figures are in the order in which they are mentioned in the main text.
2. Fig S10 was completely absent in my version - just a white square/
3. The paper starts by talking about Pro-Met-Stop, Lys-Lys-Stop etc. but then switches to PM-Stop and KK-Stop etc. Please can the authors standardise to either the 3-letter AA code or the 1-letter AA code for the whole manuscript.
4. A sentence on page 4 includes "enriched (depleted)" twice. I think "enriched/depleted" is the correct way of writing this.
5. Please add a key/table to Fig S13 writing out the COG definitions so that the casual reader can see without needing to look up what each COG class is what.
6. Could figures 2 & 3 (excel graphs) be made clearer in some way? e.g. switching to heatmaps rather than clusters of thin column bar graphs? This seems like a poor way to display the data to the reader.
7. I have to admit that I was lost in understanding how the Figure 4 work was done. I particularly struggled in working out what I was supposed to see in Panel 4C. Could this be simplified or made easier to identify for dummies like me?
8. On page 6 justification is given for doing the experimental work in *M. pneumoniae*, but is this organism a good representative of bacteria in general? Ideally it would be good to show the species-specific analysis for *M. pneumoniae* last two amino acids so we can see how it compares to the average for all bacteria and maybe compared to another species like *E.coli* for example.
9. Figure S20 is valuable but I see no description of how this analysis was done in the paper or the supplementary

10. I really liked the experimental work in the final section with ELM-Seq and Cat levels being measured. I only wish the authors also had done something in another bacteria, such as GFP in *E. coli*. I don't insist that this needs to be added for the paper to be accepted, but I think the authors would agree with me that it would be a strong addition - allowing people to see that the amino acid identity does effect expression the same way in different bacteria, and in this case in a familiar model bacterium with gene expression measured in a well understood way.

Reviewer #2:

Weber et al. report that in bacteria the final two codons, prior to the stop codon, tend to be enriched for positively charged amino acids and lacking in hydrophobic ones. The trends are most acute in highly expressed genes and only partially explained by transmembrane functionality. Experimental evidence suggests that the enrichment patterns somehow enable higher expression. The paper is well written and mostly methodologically clear.

I have several concerns about the analysis, but the greatest weakness must be the lack of mechanistic rationale to explain the effects. While the authors focus on amino acid level effects (which are unexplained), there are also codon level effects with A enrichment at the third position for reasons that also aren't clear. The paper is a helpful addition to the literature as is (pending some analysis modifications) but could be a much more valuable one with the addition of further analyses aimed at more fully dissecting the mechanistic basis (I'll not suggest experiments in this regard but leave it to the imagination of the authors). Without a better mechanistic understanding this is a relatively modest advance (see their refs 5-7, 9, 10). It will be of interest to folks working on translation dynamics, stop codon function, amino acid usage and position specific constraints. But the core message is more observational than revealing of deeper insight.

Analysis concerns:

1. I was concerned about the evidence for slow evolution of the C terminus in highly expressed genes. I have little doubt that it is true, in no small part because the best predictor of rates of protein evolution is the extent to which a gene is expressed (1). The key question is whether the positively charged sites are any more preserved than similar positively charged sites elsewhere within the same highly expressed proteins. Neither expression level nor amino acid identity are controlled in the present analysis but both are known to more generally determine rates of evolution. Finding that any amino acid is slower evolving in highly expressed genes is not news. Finding that certain transitions between amino acids are rare is also not news.
2. On a related issue, I thought that, aside from being better controlled, the evolutionary analysis could be done better. Rather than just asking about rates of evolution, one can ask whether the ancestral state predicts rates of change. If the authors are right, then if a positively charged amino acid is the ancestral state (considering two in-groups and one out-group), then the likelihood of change should be low, conversely, if the two in-group species have a non-positively charged amino acid as the ancestral state then rates of evolution should be higher at the -1 and -2 positions, than elsewhere in the same gene. This paper (2) provides a similar sort of analysis for stop codon usage.
3. I wasn't clear if the enrichment analyses control for gene level effects (I think not). Imagine that highly expressed genes tend to more commonly encode positively charged amino acids generally [in core and elsewhere]. If the probability of usage of an amino acid at site -1 is taken from core usage in all genes, then enrichment in highly expressed genes at all sites, including -1, is to be expected as a non-specific artefact. A better null then is a pairwise one in which enrichment is defined on a gene by gene basis (-1 usage v usage in the rest of that gene/core) thereby correcting for expression level, GO class etc. The data will be noisier but the analysis logic and inference cleaner. From what I can see the authors derive usage in the core - what they call the bulk - by summing usage across all proteins.
4. Alternatively, if the above analysis is too noisy, why not derive usage of each amino acid as a function of expression level (metabolically cheap amino acids tend to be used more(3) in highly expressed genes) in the core/bulk and then ask if -1 and -2 differ in usage given the expression level of the gene concerned and given this expression level specific null. If the analysis is not like this you may be relating amino acid usage skews in highly expressed genes that need have little to do with -1 or -2 proximity.

5. The same expression level control should be applied to di-amino acid usage patterns - and null should be derived from di-amino acid usage directly, not from presumed epistasis (frequency amino acid 1 * frequency of amino acid 2 = expected is not the right null as it doesn't control for bridging dinucleotide usage). It is easy to ask for all 20 x 20 amino acid pairs the expected usage given expression level.
6. In the experimental data the team look at RNA secondary structure near the stop. However, upstream of the stop in many bacterial genes are stem loop structures at the RNA level forced by RNA level palindromic repeat elements. Might the structures not be centred on the stop but upstream?
7. Codon usage trends are supposed not to be owing to codon usage trends generally as null is from the core/bulk. Again, it would be good to repeat this controlling for expression level - in either of the two approaches suggested above (pairwise noisy analysis or null derived from correlation with expression).
8. Page 4 - the AUG overlap suggestion is unclear to me - are you saying the A enrichment at nucleotide -1 is owing to gene overlap - if so exclude such genes and show the effect to goes away. I suspect it will remain as +1 overlap isn't all that common.

Minor comments

1. Page 4 - why is it interesting that threonine and lysine anti-correlate? Also emphasised on page 8, but I'm not sure why.
2. Page 4. A propos of nothing, but just as NNA-TGA can give a new start in -1 frame so it has been seen at the 5' end ATG (or NTG) followed by A gives a +1 stop and that +4A is remarkably common (4). There seems to be a most curious interplay between starts and stops. There is no need to discuss this if you don't care to, it just seems rather curious and worth bringing to your attention.
3. In the experimental part can you clarify what expression levels mean. It seems to me that the ELM-seq method reports the levels of functional DAM, not the translated levels if the amino acid changes functionally compromise the resulting protein. As the authors invented the system they should be the experts here.

1. Pal C, Papp B, Hurst LD. Highly expressed genes in yeast evolve slowly. *Genetics*. 2001;158(2):927-31.
2. Belinky F, Babenko VN, Rogozin IB, Koonin EV. Purifying and positive selection in the evolution of stop codons. *Scientific Reports*. 2018;8.
3. Akashi H, Gojobori T. Metabolic efficiency and amino acid composition in the proteomes of *Escherichia coli* and *Bacillus subtilis*. *Proceedings Of the National Academy Of Sciences Of the United States Of America*. 2002;99(6):3695-700.
4. Abrahams L, Hurst LD. Adenine Enrichment at the Fourth CDS Residue in Bacterial Genes Is Consistent with Error Proofing for +1 Frameshifts. *Mol Biol Evol*. 2017;34(12):3064-80.

Reviewer #3:

This work expands the knowledge about the C-terminal amino acid composition by analyzing 1582 genomes across the bacterial taxonomy. The authors showed a conserved bias of the last two amino acids (favoring the positively charged amino acids, while disfavoring hydrophobic amino acids and Thr), which is stronger for highly abundant proteins. They used two different reporter genes to confirm the effect of changing the C-terminal amino acids on the protein expression level in *M. pneumoniae*.

The work is in general clear and solid, but additional experiment/analysis could be performed to make the conclusions stronger:

1. The authors proposed that the effect of the C-terminal amino acid on the protein expression level could be from regulating translation termination efficiency and/or protein degradation. These two possibilities could be directly tested by (1) comparing protein degradation rate in vivo; (2) genetic analysis of measuring reporter expression in protein-degradation-deficient mutants and/or (3) measuring translation termination efficiency by ribo-seq. Apply these to some reporter mutants and also endogenous genes could elucidate whether the effect is gene-specific.

2. The authors show that the C-terminal amino acid bias is stronger for highly abundant proteins, but this is independent of codon usage or mRNA secondary structure. However, in *E. coli* at least, codon usage is highly correlated with protein abundance, and mRNA structure is highly correlated with translation efficiency. It would clarify the authors' conclusions, if they could directly compare the relationship between C-terminal amino acid composition vs all these potential determinants of protein expression level and test whether these factors are independent/epistatic in regulating the expression.

Minor points:

1. Page 3, second paragraph, in the discussion about the epistasis of the last two amino acids, "Fig 1D" should be mentioned.
2. Page 3, last paragraph, "Fig. 1D" should be "Fig. 1E".
3. Fig. S10 is blank.
4. Page 5, second paragraph, Fig. S13, Group A (RNA processing & modification), Group W (Extracellular structure) and Group Z (cytoskeleton, high Pro) also have different patterns compared to most other categories. Is there any functional impact that the authors can comment on?
5. Page 7, first paragraph, "Unexpectedly, we did not observe any significant changes in expression in the presence of proline or threonine." Why is the result of the reporter library different from the genomic analysis? Is this specific to *M. pneumoniae*?

Reviewer #1

The manuscript by Weber et al revisits the question of how variation at the protein C-terminal affects gene expression - this time focusing on bacteria and making use of the huge increase in data in the last decade thanks to genome sequencing efforts. Through a series of comparative genomics analysis methods, it is clearly shown that some amino acids are more common or rare at the gene C-terminus than random chance would allow. The analysis reveals that this is less likely to be due to codons or RNA structure and is more likely to be due to the amino acid itself. The general trends of which amino acids are enriched and depleted are seen across all bacteria (and seemingly even eukaryotes as briefly mentioned in the discussion) so it seems to be a largely universal rule. Interestingly some amino acid enrichment/depletion trends are only seen in some parts of the bacterial taxonomy, whereas others are universal. This suggests two mechanisms are going on - one universal and one more species-specific. Furthermore, the role of the protein doesn't seem to play much of a role in whether these rules are important or not.

The work also looks at how the final amino acids in proteins affect gene expression, first by their bioinformatics methods, and then using an innovative assay to assess a synthetic expression library in a Mycoplasma. This is a cool addition to an otherwise quite dry comparative genomics bioinformatics analysis paper. It provides good evidence that the amino acids chosen link to increased or decreased expression level in the cases seen in the above analysis.

Overall I'm broadly supportive of publication of this high-quality and interesting research. It focuses on work that has very broad relevance to anyone interested in bacterial gene expression (probably half of this journal's readership!). It lacks a major wow factor in terms of not ever revealing the actual mechanisms that are driving the selection of amino acid choice at the C-terminus, but that is not really needed to be honest.

We have now included a new experimental assay to measure the protein degradation rates of a reporter gene with different C-terminal amino acid identity, shedding light on one of the possible underlying mechanisms. A more detailed description of the new results can be found in the general description above and in the new results section added in the manuscript.

I recommend publication in MSB following minor corrections as suggested below.

1. The figure panel order and supplementary figure order is not in line with the main text - e.g. Fig 1D is talked about before Fig 1B. This was horribly confusing and made reading the paper very annoying. Please make sure that panels and supplementary figures are in the order in which they are mentioned in the main text.

We have changed the order of the panels in Fig. 1 such as to match the order in which they are mentioned in the main text. In addition, we have also added a few sentences to explain better the two different analyses of the amino acid pair biases.

2. Fig S10 was completely absent in my version - just a white square/

The figure S7 (previously Fig. S10) has been included back into the document.

3. The paper starts by talking about Pro-Met-Stop, Lys-Lys-Stop etc. but then switches to PM-Stop and KK-Stop etc. Please can the authors standardise to either the 3-letter AA code or the 1-letter AA code for the whole manuscript.

The abbreviations for amino acids have been standardised to the 1-letter code.

4. A sentence on page 4 includes "enriched (depleted)" twice. I think "enriched/depleted" is the correct way of writing this.

The sentence has been corrected.

5. Please add a key/table to Fig S13 writing out the COG definitions so that the casual reader can see without needing to look up what each COG class is what.

A table with the COG categories descriptions has been added to the figure EV2 (before: figure S13).

6. Could figures 2 & 3 (excel graphs) be made clearer in some way? e.g. switching to heatmaps rather than clusters of thin column bar graphs? This seems like a poor way to display the data to the reader.

We believe that in some cases, the bar graph is more adapted, as bars make it easier to directly compare the intensity of the biases for the different amino acids/codons. In the case of the biases of codons in the different stop codon contexts (Fig. 2), we have replaced the bar graph with a heatmap. In the case of the biases for amino acids in the different categories of protein abundance (Fig. 3), the differences in biases intensities are smaller and we think that the bar graph is more adapted to compare the exact values.

7. I have to admit that I was lost in understanding how the Figure 4 work was done. I particularly struggled in working out what I was supposed to see in Panel 4C. Could this be simplified or made easier to identify for dummies like me?

Figure 4 has been removed. Following the suggestions of reviewer #2, we have now completely substituted the analysis of conservation at the C-terminal by an evolutionary analysis which compares substitution rates between amino acids. We believe that the latter analysis shows in a more direct manner the existence of evolutionary forces acting at the C-terminal. We refer the reviewer to the answer to reviewer #2, questions 1-2 below.

8. On page 6 justification is given for doing the experimental work in *M. pneumoniae*, but is this organism a good representative of bacteria in general? Ideally it would be good to show the species-specific analysis for *M. pneumoniae* last two amino acids so we can see how it compares to the average for all bacteria and maybe compared to another species like *E.coli* for example.

The C-terminal biases observed (Fig. S9) in the Mollicutes class -the taxonomical clade of *M. pneumoniae*- are similar to the biases observed at the level of all bacteria, i.e. enrichment of arginine and lysine, depletion of threonine and the hydrophobic amino acids A, V, I, L, and M. In addition, it exhibits a strong depletion of proline. Thus, we believe that it is a good representative species of the biases observed in general in bacteria. We also computed the biases at the level of the single *M. pneumoniae* species (Fig. S9), and found an enrichment of lysine and depletion of threonine and proline at the C-terminal position. However, compositional biases at the finer level of a single species suffers from a lack of statistical power which prevents the detection of biases for all amino acids. The following sentence has been added to the main text at the beginning of the section "C-terminal amino acid identity impacts protein expression levels in *M. pneumoniae*":

"The C-terminal amino acid biases in this species and in the taxonomic class it belongs to, Mollicutes, are representative of the pattern of biases observed in bacteria (Appendix Fig S9)."

9. *Figure S20 is valuable but I see no description of how this analysis was done in the paper or the supplementary*

We have slightly extended the legend of the Fig. S10 (previously Fig. S20) and included some details in the Methods section “mRNA secondary structure prediction”.

10. *I really liked the experimental work in the final section with ELM-Seq and Cat levels being measured. I only wish the authors also had done something in another bacteria, such as GFP in E. coli. I don't insist that this needs to be added for the paper to be accepted, but I think the authors would agree with me that it would be a strong addition - allowing people to see that the amino acid identity does effect expression the same way in different bacteria, and in this case in a familiar model bacterium with gene expression measured in a well understood way.*

We acknowledge the fact that an additional experiment in a different bacterial species such as *E. coli*, ideally using a different reporter system, would have been a valuable contribution to the main conclusions of our study, i.e. to show that C-terminal amino acid composition may impact protein expression levels in bacteria in general. However, due to time constraints, such an experiment could not be performed. Instead, we have now included an additional experiment in *Mycoplasma pneumoniae* to measure degradation rates of the different C-terminal variants of a reporter gene, which sheds light onto the underlying mechanism that drives the observed differences in protein expression levels.

Reviewer #2

Weber et al. report that in bacteria the final two codons, prior to the stop codon, tend to be enriched for positively charged amino acids and lacking in hydrophobic ones. The trends are most acute in highly expressed genes and only partially explained by transmembrane functionality. Experimental evidence suggests that the enrichment patterns somehow enable higher expression. The paper is well written and mostly methodologically clear.

I have several concerns about the analysis, but the greatest weakness must be the lack of mechanistic rationale to explain the effects. While the authors focus on amino acid level effects (which are unexplained), there are also codon level effects with A enrichment at the third position for reasons that also aren't clear. The paper is a helpful addition to the literature as is (pending some analysis modifications) but could be a much more valuable one with the addition of further analyses aimed at more fully dissecting the mechanistic basis (I'll not suggest experiments in this regard but leave it to the imagination of the authors). Without a better mechanistic understanding this is a relatively modest advance (see their refs 5-7, 9, 10). It will be of interest to folks working on translation dynamics, stop codon function, amino acid usage and position specific constraints. But the core message is more observational than revealing of deeper insight.

We have now included a new experimental assay to measure the protein degradation rates of a reporter gene with different C-terminal amino acid identity, shedding light on one of the possible underlying mechanisms. A more detailed description of the new results can be found in the general description above and in the new results section added in the manuscript.

Analysis concerns:

1. I was concerned about the evidence for slow evolution of the C terminus in highly expressed genes. I have little doubt that it is true, in no small part because the best predictor of rates of protein evolution is the extent to which a gene is expressed (1). The key question is whether the positively charged sites are any more preserved than similar positively charged sites elsewhere within the same highly expressed proteins. Neither expression level nor amino acid identity are controlled in the present analysis but both are known to more generally determine rates of evolution. Finding that any amino acid is slower evolving in highly expressed genes is not news. Finding that certain transitions between amino acids are rare is also not news.

2. On a related issue, I thought that, aside from being better controlled, the evolutionary analysis could be done better. Rather than just asking about rates of evolution, one can ask whether the ancestral state predicts rates of change. If the authors are right, then if a positively charged amino acid is the ancestral state (considering two in-groups and one out-group), then the likelihood of change should be low, conversely, if the two in-group species have a non-positively charged amino acid as the ancestral state then rates of evolution should be higher at the -1 and -2 positions, than elsewhere in the same gene. This paper (2) provides a similar sort of analysis for stop codon usage.

[Note: As the two previous questions are related, we have taken the liberty to answer both below.]

We agree with the fact that our conservation study only addressed the key questions raised by the reviewer only superficially. We have now completely substituted our conservation study by a site-specific evolutionary analysis of the C-terminal residues, as suggested.

As the reviewer mentions, the site-specific evolutionary rates of amino acids depend on many factors: first gene-level factors such as protein expression level, then local factors such as amino acid identity, secondary structure, key residue in catalytic domain, or whether the residues are involved in a protein-protein interaction interface. Interestingly, one of the strongest determinants of the variation of evolutionary rates across sites is packing density, such that residues at the C-terminal, which are usually unstructured and/or at protein surface (at least the last 4 residues) (Jacob & Unger, 2007; Kleppe & Bornberg-Bauer, 2019), will in general have a faster evolutionary rate than residues in the core of the proteins. Indeed, purifying selection in terminal segments of proteins is usually reduced, compared to the rest of the protein (Ridout *et al*, 2010; Shabalina *et al*, 2004).

The central question we intended to answer is then whether certain amino acid substitutions were more or less frequent at the C-terminal position compared to other similar positions in the proteins. More precisely, we hypothesized that if the ancestral state is a favorable amino acid (positively charged), purifying selection would decrease the substitution rate to non-favorable amino acids. Contrariwise, if the ancestral state is a non-favorable amino acid (hydrophobic amino acids and threonine), positive selection would increase the substitution rate to favorable amino acids. Ideally, one would compare the C-terminal position to other positions in the same protein that are equally exposed to the solvent, which would require knowledge of the protein three-dimensional structure. As the solvent accessibility of residues is particularly high at the last 2 positions of the C-terminal, we reasoned that the -2 position would be an appropriate site for a null model, with similar evolutionary constraints than the position -1.

A new results section, "Pattern of amino acid substitution rates suggests C-terminal-specific purifying and positive selections", has been included in the main text, with its corresponding methods section. Our results suggest the existence of positive selection for positively charged amino acids and purifying selection against threonine that are specific to the C-terminal -1 position. Without repeating the results already described in the main text, we would like to comment further some of the aspects of the analysis.

Apart from the analyses described in the main text, we also tried to combine the protein abundance data from paxDB to the selected triplets of genomes, in order to study the substitution rates in highly abundant and lowly abundant proteins (see additional methods below). However, the number of proteins and substitution events at the C-terminal position resulted too small to allow for reliable statistics of substitution rates (6355 protein alignments with abundance value).

Evolutionary analysis, additional methods:

Protein abundance data sets were downloaded from the paxDB database release 4.1 (Wang *et al*, 2015). We selected 6 species in paxDB (*Listeria monocytogenes* EGD-e, *Shewanella oneidensis* MR-1, *Bacillus subtilis* subsp. *subtilis* str. 168, *Bacillus anthracis* str. Sterne, *Helicobacter pylori* 26695, *Bartonella henselae* str. Houston-1) which were closely related to one of the species triplets at the taxonomic level. More precisely, each of the selected species belonged to the same clade at the species rank of at least one genome in the list of species triplets. Protein abundance information from the paxDB was propagated to its closest homolog in the corresponding triplet species. Homologs were identified by searching for protein sequence with at least 90% full-length identity ($\text{alignment identity} \times \text{alignment length} / \text{query protein length}$) using BLASTp (Altschul *et al*, 1997). In the case of multiple hits (120 cases) or multiple query proteins for the same target (11 cases), the protein with highest full-length identity was chosen. In total, 8208 proteins out of 9971 from paxDB were matched to a homologous protein in the corresponding triplet species. The coverage in terms of number of proteins with propagated abundance values in each of the 6 triplet species ranged from 16% for *Listeria monocytogenes* (GCF_000438605.1) to 73% for *Bacillus subtilis* (GCF_000959025.1). We then considered the abundance value of the ATGC protein representative of the abundance of the ATGC-COG it belonged to. We thus obtained 6355 ATGC-COGs with an assigned abundance value. The ATGC-COGs abundance values were binned into three percentile ranges: 0-25% (low abundance), 25%-75% (medium abundance), and 75%-100% (high abundance). Each of the extreme bins contained 1662 ATGC-COGs. This number of protein sequence alignments resulted too small to allow reliable estimation of site-specific amino acid substitution rates, even when considering transitions between groups of amino acids. For example, 351 protein triplets presented an ancestral state with a positively charged amino acid (K, R) at the C-terminal position -1, of which only 8 substitution events to a different amino acid group could be identified.

3. I wasn't clear if the enrichment analyses control for gene level effects (I think not). Imagine that highly expressed genes tend to more commonly encode positively charged amino acids generally [in core and elsewhere]. If the probability of usage of an amino acid at site -1 is taken from core usage in all genes, then enrichment in highly expressed genes at all sites, including -1, is to be expected as a non-specific artefact. A better null then is a pairwise one in which enrichment is defined on a gene by gene basis (-1 usage v usage in the rest of that gene/core) thereby correcting for expression level, GO class etc. The data will be noisier but the analysis logic and inference cleaner. From what I can see the authors derive usage in the core - what they call the bulk - by summing usage across all proteins.

4. Alternatively, if the above analysis is too noisy, why not derive usage of each amino acid as a function of expression level (metabolically cheap amino acids tend to be used more(3) in highly expressed genes) in the core/bulk and then ask if -1 and -2 differ in usage given the expression level of the gene concerned and given this expression level specific null. If the analysis is not like this you may be relating amino acid usage skews in highly expressed genes that need have little to do with -1 or -2 proximity.

[Remark: We copy below a similar question raised by reviewer #3 and provide a unique answer to all three questions.]

reviewer #3, question 2. The authors show that the C-terminal amino acid bias is stronger for highly abundant proteins, but this is independent of codon usage or mRNA secondary structure. However, in E. coli at least, codon usage is highly correlated with protein abundance, and mRNA structure is highly correlated with translation efficiency. It would clarify the authors' conclusions, if they could directly compare the relationship between C-terminal amino acid composition vs all these potential determinants of protein expression level and test whether these factors are independent/epistatic in regulating the expression.

In the analysis of biases in high, medium and low abundance protein groups, we always used as a background the frequencies of amino acid in the bulk of all the proteins *in the same group*. Indeed, the amino acid frequencies in the bulk of highly abundant proteins group were different from the ones in the lowly abundant proteins group (Fig. S8). In the highly abundant protein group, K is found in the bulk at a frequency of 5.92%. Frequency of K at C-terminal position -1 was 15.5%, and the odds ratio, calculated as the unconditional maximum likelihood estimate of the Fisher test, was 2.91. In this case, the frequency of K at position -1 was significantly higher than the frequency of K in the bulk of highly abundant proteins. Correspondingly, in the lowly abundant protein group, K is found in the bulk at a frequency of 4.19%. Frequency of K at C-terminal position -1 was 8.31%, and the odds ratio was 2.07. In this case, the frequency of K at position -1 was also significantly higher than the frequency of K in the bulk of lowly abundant proteins. Even if the frequency of K in the bulk varies considerably between the two protein groups, we still find K enriched at the C-terminal in each group. Thus, the enrichment of K at the C-terminal of highly and lowly abundant proteins is not an artefact of an inappropriate background model of amino acid usages. However, the odds ratio, or the amplitude of the enrichment, is clearly smaller in the case of the lowly abundant protein group compared to the highly abundant protein group. This difference suggests that the evolutionary forces that drive the preference for K at the C-terminal are stronger in the highly abundant proteins. We have

clarified our approach in the main text in the results section “C-terminal amino acid identity is associated with protein abundance”.

As the reviewers mentioned, comparing amino acid usages on a gene-by-gene basis would correct for any factor influencing amino acid composition at the gene level, such as expression level, functional category, secondary structure of mRNA, etc. Such an approach would certainly improve the null model. However, it would also reduce the size of the observation to a single residue at the C-terminal specific site, which renders this approach impractical from a statistical point of view. In this sense, no specific preference or amino acid usage at a specific position of the C-terminal can be defined at the level of a single protein.

Thus, in our study, we followed an inherently coarse-grain rationale by considering groups of proteins, such that trends in amino acid frequency at a specific C-terminal position could be compared to the frequency in the bulk positions of the protein sequences. In order to correct for different potentially confounding factors, we grouped proteins into bins of proteins with similar values of the factor, e.g. abundance. In each group, we used as a null model the amino acid usage in the bulk of the proteins in the group. In such a way, we effectively corrected for the potential effects of the factor on amino acid usage, albeit at a coarse-grain level. One advantage of this approach, is that the size and the number of the bins can be modulated, such as to find a trade-off between the accuracy of the correction for the factor (correct amino acid usage in the bulk for a specific level of abundance) and the number of observations (how many amino acids we observe at the C-terminal position -1) which determines the statistical power of the comparison.

Overall, we emphasize that the aim of our analysis was to show that *some difference* in the intensity of the biases exists between the lowly and highly abundant proteins. We focused on the lowest (0-20%) and highest (80-100%) bins of protein abundances, to obtain an accurate enough bulk amino acid usage (null model) and enough observations of C-terminal amino acids, such as to obtain reliable comparisons. As a control, we have also performed the same analysis using 20 smaller bins of protein abundance (Fig. S101 below), by steps of 5 percentiles. In this case, we observed the same trend in the intensity of the biases between the lowest abundance bins and the highest abundance bins, showing that the results of the larger bins captured correctly the amino acid usages. Notice that in this case, the number of proteins in each bin is much smaller, which leads to some variability in the estimation of biases.

Protein abundance is determined by multiple gene-level factors, such as translation efficiency, mRNA secondary structure at various positions along the transcript, codon usage, etc. Deciphering the exact contribution of all these factors to protein abundance and their relationship to C-terminal amino acid composition would require a much larger dataset. First, based on the fold changes observed in our experimental assay, we believe that C-terminal amino acid composition is not the strongest determinant of protein abundance, but rather a weak determinant. Thus, most of the other known features, as for example codon usage or strength of the ribosome binding site, can be expected to have a stronger impact on protein abundance. Studying the contribution of C-terminal amino acid composition to protein abundance among the other features would thus require an even larger amount of sequences with the corresponding experimental data. If such data were available, one possible approach would be to apply machine learning method to predict protein abundance based on gene features, and evaluate the predictive power of C-terminal amino acids identity.

In our study, we aimed at identifying and isolating the specific contribution of C-terminal amino

acid composition on protein expression. First, at the large-scale level of the genomic analysis of bacterial species, we observed general trends in the preferences of C-terminal amino acids. In this first study, we aimed at averaging out differences across species, and found a general association between protein abundance and the intensity of the biases. Second, in our ELM-seq experimental assay, we systematically varied the last two codons and analyzed the specific contribution of C-terminal amino acids to protein abundance in a controlled experimental design. In this case, the differences in protein expression levels can only be attributed to the C-terminal sequence variation.

Fig. S101

Fig. S101. Protein abundance and C-terminal amino acid identity, additional analysis.

Proteins were categorized into bins of abundances, following percentile ranges in step of 5 percentiles. C-terminal amino acid composition biases at position -1, with respect to the bulk frequency of each category, were analyzed for each of the categories. Enrichment of lysine clearly increases with protein abundance group, and bias against threonine becomes stronger.

*5. The same expression level control should be applied to di-amino acid usage patterns - and null should be derived from di-amino acid usage directly, not from presumed epistasis (frequency amino acid 1 * frequency of amino acid 2 = expected is not the right null as it doesn't control for bridging dinucleotide usage). It is easy to ask for all 20 x 20 amino acid pairs the expected usage given expression level.*

In the case of the bias analysis in the bacterial kingdom, we compared the frequency of amino acid pairs at the C-terminal in two ways: first, compared to the di-amino acid usage in the bulk (Fig. 1B), and second, to the expected frequency at position -1 and -2 (epistasis, Fig. 1C). We have clarified this part of the analysis in the main text, see results section “C-terminal amino acid and codon composition in bacteria is biased”. In the cases of the bias analysis in highly abundant proteins, the total number of proteins with abundance data was much lower, which greatly limited the statistical power for the amino acid pair analysis (data not shown).

6. In the experimental data the team look at RNA secondary structure near the stop. However, upstream of the stop in many bacterial genes are stem loop structures at the RNA level forced by RNA level palindromic repeat elements. Might the structures not be centred on the stop but upstream?

In order to take into account possible secondary structure upstream of the stop codon, we repeated the same analysis by computing RNA folding energy in a window of 54 nt spanning 30 nucleotides upstream the randomized region, to 15 nucleotides downstream the stop codon. This window was large enough to include a GC-rich hexamer upstream of the randomized region that could potentially form hairpins with the randomized region:

AAACCAGGAGTCGTTTCACCCGCGAAAAAANNNNNNTAANNNNNNCAGGCCTTGAAGATA

We classified sequence variants into 4 bins of folding energy and observed no significant differences in the expression levels, neither in the weak nor in the strong promoter library. Thus, while some strong secondary structures could potentially appear in certain variants, they do not correlate with the measured protein expression levels.

We further reasoned that, if the main effect on protein expression was due to the identity of the C-terminal amino acid, the effect of secondary structure could still be observed when comparing synonymous codons. However, we found that the relative change in expression levels among synonymous codons was not correlated to the folding energy.

7. Codon usage trends are supposed not to be owing to codon usage trends generally as null is from the core/bulk. Again, it would be good to repeat this controlling for expression level - in either of the two approaches suggested above (pairwise noisy analysis or null derived from correlation with expression).

In the case of codon biases, we followed the same approach of grouping proteins into bins of low abundance and high abundance, for the reasons exposed above. However, the number of sequences available in each of the abundance bin resulted too small to obtain reliable statistics of codon biases. In order to reach the same statistical power, more protein sequences are needed to analyze codons biases (64 states) than amino acid biases (20 states).

8. Page 4 - the AUG overlap suggestion is unclear to me - are you saying the A enrichment at nucleotide -1 is owing to gene overlap - if so exclude such genes and show the effect to goes away. I suspect it will remain as +1 overlap isn't all that common.

Indeed, this was our hypothesis, that the preference for NNA codons was due to the overlapping

of start codons. Following reviewer's suggestion, we have computed the codon biases for genes in the UGA stop codon context by excluding the cases where the start codon of the annotated downstream gene was overlapping the stop codon at nucleotide position -1, e.g. NNA-UGA where AUG is the downstream start codon. These cases were in fact fairly frequent, with 16.6% of all genes ending with UGA stop codon having an overlapping start codon at position -1 in our set of 1582 bacterial genomes. We found that the specific preference for NNA codons was greatly reduced, down to non-significance, when excluding these cases. This result supports the view that the preference for NNA codons in the UGA stop codon context is mainly due to the overlapping of start codons. In this respect, we have updated Figure 2 by including the biases for the UGA context with excluded start codons, and have extended the main text at the end of the section "Pattern of C-terminal codon biases and its relationship to the stop codon context".

Minor comments

1. Page 4 - why is it interesting that threonine and lysine anti-correlate? Also emphasised on page 8, but I'm not sure why.

The observed strong anticorrelation across all phyla between lysine and threonine biases could point to a common mechanism producing these two preferences. However, our experimental results showed that C-terminal threonine had no influence on protein expression level or degradation rate. While these assays were performed in only one species, they suggest that the bias against threonine observed in bacteria is due to a different mechanism than the biases in favor of lysine. Further investigations should be performed in order to reveal the mechanism that drives the selection against threonine at the C-terminal.

2. Page 4. A propos of nothing, but just as NNA-TGA can give a new start in -1 frame so it has been seen at the 5' end ATG (or NTG) followed by A gives a +1 stop and that +4A is remarkably common (4). There seems to be a most curious interplay between starts and stops. There is no need to discuss this if you don't care to, it just seems rather curious and worth bringing to your attention.

We were not aware of the frameshift correction model for the enrichment of +4 adenine after start codon. Interestingly, the authors of (Abrahams & Hurst, 2017) also tested the effect of the overlapping of start and stop codons, and found that the +4A was still enriched in genes with no overlap. Apart from the complexity of the multiple selective pressures and their related mechanisms that shape the N-terminal of coding sequences, we find that the evolutionary rules that govern gene overlap are surprisingly complex. It has been suggested that gene overlap is, in general, driven by multiple mechanisms. For example, genome minimization may provide a direct benefit by increasing rates of replication (Rogozin *et al*, 2002). Among overlapping genes, the overlap length of 4 bp is remarkably common (Lillo & Krakauer, 2007). As discussed by Lillo and Krakauer, one of the mechanisms that could explain the preponderance of short overlaps in operons could be translational coupling. A recent study suggests that translational coupling could be widespread in prokaryotes, and provides some experimental evidence for it (Huber *et*

al, 2019). Short overlaps of 4 bp could be particularly efficient to promote the “termination-reinitiation” mechanism, where the same ribosome, or at least the small subunit, could directly initiate translation of a nearby start codon shortly after termination, without unbinding from the mRNA. Whether such a mechanism is common in bacteria remains under debate. We believe that the mechanism of translational coupling is an attractive hypothesis to explain the high occurrence of 4 bp overlaps, and that it could play an essential role in providing efficient coordination of gene regulation.

Interestingly, *M. pneumoniae* is a bacterial species that has undergone genome reduction (Razin *et al*, 1998; Meseguer *et al*, 2003), and has reassigned UGA from stop codon to tryptophan, a feature found in most mollicutes (Sirand-Pugnet *et al*, 2007). Therefore, in this bacterium, the 4 bp overlap between UGA stop codon and a start codon cannot be used. Nevertheless, we observe that the 1 bp overlap UAAUG is very common, with 55 out of 715 genes exhibiting this kind of overlap with their downstream neighbour. It has been argued that this heavy use of gene overlapping does not apparently result from genome compaction (Razin *et al*, 1998). Due to its basic regulatory machinery, both at the level of transcription (Yus *et al*, 2019) and translation (Grosjean *et al*, 2014), and the fact that the Shine-Dalgarno motif has little effect on protein expression levels in this organism (Yus *et al*, 2017), we speculate that translational coupling could be particularly relevant in this bacterium and could partially explain the high degree of gene overlapping.

3. In the experimental part can you clarify what expression levels mean. It seems to me that the ELM-seq method reports the levels of functional DAM, not the translated levels if the amino acid changes functionally compromise the resulting protein. As the authors invented the system they should be the experts here.

The ELM-seq method measures the level of activity of the *E. coli* Dam methylase, by quantifying the relative level of DNA methylation at the 4 GATC sites on the reporter cassette. The authors of the original method (Yus *et al*, 2017) showed that in the case of the unmodified Dam protein sequence, the methylase activity is proportional to the protein level (as measured by mass spectrometry). In our experimental study, we made the assumption that the activity of the Dam protein is not affected by the identity of the fused C-terminal residues. Thus, the observed differences in methylation levels must reflect differences in protein expression levels. We have clarified this assumption in the main text. We believe that such an assumption is reasonable in the case of the Dam protein. The crystallographic structure of the unmodified *E. coli* Dam DNA-(adenine-N6)-methyltransferase in complex with cognate DNA and in the presence of S-adenosyl-l-homocysteine (Horton *et al*, 2006) shows that the last 8 C-terminal residues (271–278) are disordered. Moreover, the C-terminal region lies far away from both the DNA binding domain and the substrate binding pocket. The residue of the resolved structure closest to the C-terminal, P270, lies at a distance >20 Å from both the DNA molecule and the ligand. Although we cannot strictly exclude such a possibility, it is unlikely that the extension of two amino acids at the C-terminal would affect the activity of the Dam protein.

In the case of the new luciferase experimental assay, we similarly assume that the C-terminal extension did not affect the luciferase activity, and that activity was proportional to the abundance of the luciferase protein. Of the residues identified so far that play a role in the

luciferase catalytic reaction (Modestova *et al*, 2014), none are located in the C-terminal region (last 6 disordered residues).

Finally, similar trends were found in the expression levels of the C-terminal variants in the *cat* reporter assay. In this case, protein abundance was directly measured by an ELISA based assay.

Although we cannot rule out a potential influence of the C-terminal extensions on the protein activity, that fact that qualitatively similar results are obtained with three different gene reporters supports the assumption that the C-terminal extension did not affect protein activity, and thus that protein activity can be treated as a readout for protein abundance.

1. Pal C, Papp B, Hurst LD. *Highly expressed genes in yeast evolve slowly. Genetics.* 2001;158(2):927-31.

2. Belinky F, Babenko VN, Rogozin IB, Koonin EV. *Purifying and positive selection in the evolution of stop codons. Scientific Reports.* 2018;8.

3. Akashi H, Gojobori T. *Metabolic efficiency and amino acid composition in the proteomes of Escherichia coli and Bacillus subtilis. Proceedings Of the National Academy Of Sciences Of the United States Of America.* 2002;99(6):3695-700.

4. Abrahams L, Hurst LD. *Adenine Enrichment at the Fourth CDS Residue in Bacterial Genes Is Consistent with Error Proofing for +1 Frameshifts. Mol Biol Evol.* 2017;34(12):3064-80.

Reviewer #3

*This work expands the knowledge about the C-terminal amino acid composition by analyzing 1582 genomes across the bacterial taxonomy. The authors showed a conserved bias of the last two amino acids (favoring the positively charged amino acids, while disfavoring hydrophobic amino acids and Thr), which is stronger for highly abundant proteins. They used two different reporter genes to confirm the effect of changing the C-terminal amino acids on the protein expression level in *M. pneumoniae*.*

The work is in general clear and solid, but additional experiment/analysis could be performed to make the conclusions stronger:

1. The authors proposed that the effect of the C-terminal amino acid on the protein expression level could be from regulating translation termination efficiency and/or protein degradation. These two possibilities could be directly tested by (1) comparing protein degradation rate in vivo; (2) genetic analysis of measuring reporter expression in protein-degradation-deficient mutants and/or (3) measuring translation termination efficiency by ribo-seq. Apply these to some reporter mutants and also endogenous genes could elucidate whether the effect is gene-specific.

We have now included a new experimental assay to measure the protein degradation rates of a reporter gene with different C-terminal amino acid identity, shedding light on the underlying mechanism. A more detailed description of the new results can be found in the general description above and in the new results section added to the main text. These results show that approximately 85% of the observed variance in the protein expression levels could be explained by the differences in protein degradation rate between C-terminal variants of a reporter gene, suggesting the protein degradation is one of the main mechanisms driving the effect of C-terminal amino acid on protein expression level.

*2. The authors show that the C-terminal amino acid bias is stronger for highly abundant proteins, but this is independent of codon usage or mRNA secondary structure. However, in *E. coli* at least, codon usage is highly correlated with protein abundance, and mRNA structure is highly correlated with translation efficiency. It would clarify the authors' conclusions, if they could directly compare the relationship between C-terminal amino acid composition vs all these potential determinants of protein expression level and test whether these factors are independent/epistatic in regulating the expression.*

As the reviewer #2 raised a very similar question, we have taken the liberty to answer the two questions together. Please refer to the answer to questions 3 and 4 of reviewer #2.

Minor points:

1. Page 3, second paragraph, in the discussion about the epistasis of the last two amino acids, "Fig 1D" should be mentioned.

The figure is now mentioned.

2. Page 3, last paragraph, "Fig. 1D" should be "Fig. 1E".

The error has been corrected.

3. Fig. S10 is blank.

The figure S7 (previously Fig. S10) has been included back into the document.

4. Page 5, second paragraph, Fig. S13, Group A (RNA processing & modification), Group W (Extracellular structure) and Group Z (cytoskeleton, high Pro) also have different patterns compared to most other categories. Is there any functional impact that the authors can comment on?

For the COG categories A, B, W and Z, we observe slightly different compositional biases at the C-terminal, although lysine is still favored and threonine disfavored. The biggest differences with the main pattern are: in categories W and Z, depletion of D, S, G, A, and enrichment of F; in category W, enrichment of W. Although we could not relate these differences to specific characteristics of the proteins in these functional categories, it is worth noting that these categories contain the lowest number of proteins (from 396 to 1490, compared to 242 738 in the J category). As such, their compositional biases might be more dependent on the C-terminal composition of a few specific orthologous groups. We have slightly modified the legend of the Figure EV2 (before: figure S13) to reflect this fact.

5. Page 7, first paragraph, "Unexpectedly, we did not observe any significant changes in expression in the presence of proline or threonine." Why is the result of the reporter library different from the genomic analysis? Is this specific to M. pneumoniae?

The biases for C-terminal amino acids in the genomic analysis most likely reflect a combination of multiple evolutionary pressures. As mentioned in the discussion, one possible interpretation could be that the presence of threonine or proline impacts the general fitness of the organism without directly reducing protein expression levels. If this is the case, the presence of threonine and proline would be selected against at the C-terminal of proteins in bacterial species, but no change would be observed in a protein expression assay. Another possibility could be that the presence of threonine or proline reduces protein expression in a species-dependent manner, with no effect in *M. pneumoniae*.

References

- Abrahams L & Hurst LD (2017) Adenine enrichment at the fourth CDS residue in bacterial genes is consistent with error proofing for+ 1 frameshifts. *Mol. Biol. Evol.* **34**: 3064–3080
- Altschul SF, Madden TL, Schäffer AA, Zhang J, Zhang Z, Miller W & Lipman DJ (1997) Gapped BLAST and PSI-BLAST: a new generation of protein database search programs. *Nucleic Acids Res.* **25**: 3389–3402
- Grosjean H, Breton M, Sirand-Pugnet P, Tardy F, Thiaucourt F, Citti C, Barré A, Yoshizawa S, Fourmy D, de Crécy-Lagard V & Blanchard A (2014) Predicting the minimal translation apparatus: lessons from the reductive evolution of mollicutes. *PLoS Genet.* **10**: e1004363
- Horton JR, Liebert K, Bekes M, Jeltsch A & Cheng X (2006) Structure and substrate recognition of the Escherichia coli DNA adenine methyltransferase. *J. Mol. Biol.* **358**: 559–570
- Huber M, Faure G, Laass S, Kolbe E, Seitz K, Wehrheim C, Wolf YI, Koonin EV & Soppa J (2019) Translational coupling via termination-reinitiation in archaea and bacteria. *Nat. Commun.* **10**: 4006
- Jacob E & Unger R (2007) A tale of two tails: why are terminal residues of proteins exposed? *Bioinformatics* **23**: e225–e230 Available at: <http://dx.doi.org/10.1093/bioinformatics/btl318>
- Kleppe AS & Bornberg-Bauer E (2019) Robustness by intrinsically disordered C-termini and translational readthrough. *Nucleic Acids Res.* Available at: <http://dx.doi.org/10.1093/nar/gkz1106>
- Lillo F & Krakauer DC (2007) A statistical analysis of the three-fold evolution of genomic compression through frame overlaps in prokaryotes. *Biol. Direct* **2**: 22
- Meseguer MA, Alvarez A, Rejas MT, Sánchez C, Pérez-Díaz JC & Baquero F (2003) Mycoplasma pneumoniae: a reduced-genome intracellular bacterial pathogen. *Infect. Genet. Evol.* **3**: 47–55
- Modestova Y, Koksharov MI & Ugarova NN (2014) Point mutations in firefly luciferase C-domain demonstrate its significance in green color of bioluminescence. *Biochim. Biophys. Acta* **1844**: 1463–1471
- Razin S, Yogev D & Naot Y (1998) Molecular biology and pathogenicity of mycoplasmas. *Microbiol. Mol. Biol. Rev.* **62**: 1094–1156
- Ridout KE, Dixon CJ & Filatov DA (2010) Positive selection differs between protein secondary structure elements in Drosophila. *Genome Biol. Evol.* **2**: 166–179
- Rogozin IB, Spiridonov AN, Sorokin AV, Wolf YI, Jordan IK, Tatusov RL & Koonin EV (2002) Purifying and directional selection in overlapping prokaryotic genes. *Trends Genet.* **18**: 228–232
- Shabalina SA, Ogurtsov AY, Rogozin IB, Koonin EV & Lipman DJ (2004) Comparative analysis of orthologous eukaryotic mRNAs: potential hidden functional signals. *Nucleic Acids Res.*

32: 1774–1782

Sirand-Pugnet P, Citti C, Barré A & Blanchard A (2007) Evolution of mollicutes: down a bumpy road with twists and turns. *Res. Microbiol.* **158**: 754–766

Wang M, Herrmann CJ, Simonovic M, Szklarczyk D & von Mering C (2015) Version 4.0 of PaxDb: Protein abundance data, integrated across model organisms, tissues, and cell-lines. *Proteomics* **15**: 3163–3168

Yus E, Lloréns-Rico V, Martínez S, Gallo C, Eilers H, Blötz C, Stülke J, Lluch-Senar M & Serrano L (2019) Determination of the Gene Regulatory Network of a Genome-Reduced Bacterium Highlights Alternative Regulation Independent of Transcription Factors. *Cell Systems* **9**: 143–158.e13 Available at: <http://dx.doi.org/10.1016/j.cels.2019.07.001>

Yus E, Yang J-S, Sagues A & Serrano L (2017) A reporter system coupled with high-throughput sequencing unveils key bacterial transcription and translation determinants. *Nat. Commun.* **8**: 368

2nd Editorial Decision

26th March 2020

Thank you for sending us your revised manuscript. We have now heard back from the reviewer who was asked to evaluate your study. As you will see below, the reviewer is satisfied with the modifications made and is supportive of publication.

Before we formally accept the manuscript for publication, we would ask you to address a few remaining editorial issues listed below.

REFEREE REPORTS

Reviewer #2:

I very much enjoyed the first version of this paper but had a few concerns regarding some of the statistical analyses and very much wanted more insight into the underlying causes of the trends. The authors have done a good job on both fronts. The new species triplet analysis is an improvement on what was before. The clarification that null models for enrichment by expression class had the null specified for the expression class is important to know. I'm happy that the DAM system is likely not influenced directly by the changes made so the assumption that activity = level seems valid (but probably not definitive). The new data showing that the effect is mediated largely through differential protein degradation rates, with the possible involvement of proteases, is both helpful and strongly suggests the next experiments (I'll refrain from the usual request for ever more experiments - there is more than enough here for a substantial and interesting paper).

I have no further concerns. Nice work.

2nd Revision - authors' response

7th April 2020

The Authors have made the requested editorial changes.

Accepted

9th April 2020

Thank you again for sending us your revised manuscript. We are now satisfied with the modifications made and I am pleased to inform you that your paper has been accepted for publication.

Corresponding Author Name: Luis Serrano
Journal Submitted to: Molecular Systems Biology
Manuscript Number: MSB-19-9208R